# miRNA-mediated control of gephyrin synthesis drives sustained inhibitory synaptic plasticity

Theresa M Welle [1,2], Dipen Rajgor [1,2], Dean J Kareemo [1], Joshua D Garcia [1], Sarah M Zych [1], Sarah E Wolfe [1], Sara E Gookin [1], Tyler P Martinez [1], Mark L Dell'Acqua [1], Christopher P Ford [1], Matthew J Kennedy [1] & Katharine R Smith [1]✉

## Abstract

**Activity-dependent protein synthesis is crucial for long-lasting forms of synaptic plasticity. However, our understanding of translational mechanisms controlling GABAergic synapses is limited. One distinct form of inhibitory long-term potentiation (iLTP) enhances postsynaptic clusters of GABA_ARs and the primary inhibitory scaffold, gephyrin, to promote sustained synaptic strengthening. While we previously found that persistent iLTP requires mRNA translation, the mechanisms controlling plasticity-induced gephyrin translation remain unknown. We identify miR153 as a novel regulator of *Gphn* mRNA translation which controls gephyrin protein levels and synaptic clustering, ultimately impacting inhibitory synaptic structure and function. iLTP induction downregulates miR153, reversing its translational suppression of *Gphn* mRNA and promoting de novo gephyrin protein synthesis and synaptic clustering during iLTP. Finally, we find that reduced miR153 expression during iLTP is driven by an excitation-transcription coupling pathway involving calcineurin, NFAT and HDACs, which also controls the miRNA-dependent upregulation of GABA_ARs. Together, we delineate a miRNA-dependent post-transcriptional mechanism that controls the expression of the key synaptic scaffold, gephyrin, and may converge with parallel miRNA pathways to coordinate gene upregulation to maintain inhibitory synaptic plasticity.**

**Keywords** GABAA Receptor; Inhibitory Synapse; Gephyrin; Translation; miRNA
**Subject Categories** Neuroscience; RNA Biology; Translation & Protein Quality

## Introduction

Activity-dependent, long-term changes in excitatory and inhibitory synaptic strength are crucial for tuning neural excitability, sculpting networks, and supporting higher order brain functions involved in learning, memory, and cognition (Akhondzadeh, 1999; Takeuchi et al, 2013). Synaptic function and plasticity are supported by continuous changes in gene expression via processes like transcription and translation to alter the synaptic proteome. Indeed, mRNA translation is essential for many different forms of plasticity and particularly critical for shaping synaptic strength long-term (Martin et al, 1997; Klann et al, 2004; Sutton and Schuman, 2006; Gal-Ben-Ari et al, 2012; Sidrauski et al, 2013; Rosenberg et al, 2014; Laguesse and Ron, 2020; Rajgor et al, 2020; Rajgor et al, 2021). Although there is a wealth of research uncovering how translation facilitates long-term plasticity of glutamatergic synapses (Costa-Mattioli et al, 2009; Cajigas et al, 2010; Rajgor et al, 2021), our understanding of its contribution to activity-dependent changes in GABAergic synaptic strength remains limited. Synaptic inhibition modulates excitatory plasticity and circuit firing, maintaining an appropriate level of neural activity via excitatory/inhibitory (E/I) balance (Klausberger and Somogyi, 2008; Chiu et al, 2019). Therefore, understanding how activity-dependent changes in gene expression contribute to long-term inhibitory synaptic strength is imperative for a complete view of brain function.

In the central nervous system, synaptic inhibition is primarily mediated by GABAergic synapses, where GABA type A receptors (GABA_ARs) are clustered opposite GABA-releasing presynaptic terminals by the principal scaffold, gephyrin (Fritschy et al, 2008; Tretter et al, 2008; Mukherjee et al, 2011; Tretter et al, 2011; Tyagarajan and Fritschy, 2014; Lorenz-Guertin and Jacob, 2018). Gephyrin plays a crucial role in GABAergic synaptic function and plasticity, and its disruption is associated with neuropathologies including epilepsy and Alzheimer's disease (Agarwal et al, 2008; Smith and Kittler, 2010; Fang et al, 2011; González, 2013; Hales et al, 2013; Dejanovic et al, 2014; Kiss et al, 2016; Dejanovic et al, 2015; Mele et al, 2019). GABAergic synapses can undergo numerous forms of bidirectional plasticity to strengthen or weaken synaptic inhibition, thus impacting neuronal excitability (Bar-Ilan et al, 2013; Bloss et al, 2016; Barberis, 2020). An important form of plasticity is inhibitory long-term potentiation (iLTP), which is mediated by NMDA receptor (NMDAR) activity (Marsden et al, 2007; Marsden et al, 2010; Petrini et al, 2014; Chiu et al, 2018; Rajgor et al, 2020), driving heterosynaptic changes at GABAergic postsynapses. Given the importance of iLTP for controlling excitatory synaptic potentiation (Leão et al, 2012; Williams and Holtmaat, 2019; Udakis et al, 2020) and its role in learning and sensory experience in vivo (Kannan et al, 2016; Udakis et al, 2020),

[1]Department of Pharmacology, University of Colorado School of Medicine, Anschutz Medical Campus, 12800 East 19th Avenue, Aurora, CO 80045, USA. [2]These authors contributed equally: Theresa M Welle, Dipen Rajgor. ✉E-mail: katharine.r.smith@cuanschutz.edu

it is crucial to understand the underlying mechanisms driving this process.

Previous work found that the strengthening of inhibitory synapses during iLTP is primarily driven by increased postsynaptic clustering of GABA$_A$Rs and gephyrin (Marsden et al, 2007; Marsden et al, 2010; Petrini et al, 2014; Rajor et al, 2020). In the first 20-30 min, this clustering is accommodated by recruitment of pre-existing proteins (Petrini et al, 2014; Rajor et al, 2020). However, we recently showed that to sustain iLTP over longer timescales, translation of GABAergic synaptic components is required (Rajor et al, 2020). Following iLTP induction, translational repression of *Gabra1* and *Gabrg2* mRNAs, encoding synaptic GABA$_A$R subunits α1 and γ2, is relieved to enable de novo synthesis and increased GABA$_A$R synaptic clustering. We found that this mechanism is mediated by the microRNA, miR376c. MicroRNAs (miRNA) are key regulators of activity-dependent translation in neurons, where they control expression of numerous synaptic genes (Smalheiser and Lugli, 2009; Im and Kenny, 2012; Hu and Li, 2017; Reza-Zaldivar et al, 2020). However, the mechanisms driving translation of gephyrin, the critical inhibitory scaffold, remain unknown. In this work, we aimed to characterize the mechanisms which control gephyrin translation and its upregulation during iLTP.

Here we identify miR153 as a novel regulator of gephyrin mRNA (*Gphn*) translation in neurons and find that its downregulation is critical for sustaining GABAergic synaptic clustering during iLTP. Following iLTP stimulation, miR153 expression is reduced, allowing *Gphn* translation, and the subsequent increase of gephyrin and GABA$_A$R clusters at inhibitory synapses. Remarkably, we find that miR153 is transcriptionally repressed following iLTP stimulation, a process that is controlled by the same signaling pathway regulating miR376c expression during iLTP. Together, our findings provide a mechanism by which iLTP stimulation can induce signaling to coordinate the downregulation of miR376c and miR153 concurrently, thereby modulating the expression of their targets (*Gabra1*, *Gabrg2* and *Gphn*), which are essential for potentiating synaptic inhibition.

## Results

### miR153 regulates *Gphn* translation during iLTP

To investigate whether gephyrin mRNA translation could be controlled by miRNA activity during iLTP, we treated cultured hippocampal neurons with an extensively characterized iLTP protocol (20 μM NMDA, 10 μM CNQX for 2 min) which augments inhibitory synaptic transmission (Marsden et al, 2007; Marsden et al, 2010; Petrini et al, 2014; Rajor et al, 2020), and is sustained for at least 90 min post-stimulation (Rajor et al, 2020). At 90 min following iLTP or sham stimulation (Ctrl) we performed Argonaute 2 (AGO2) immunoprecipitation assays. AGO2 is part of the RNA-Induced Silencing Complex (RISC), which is crucial for miRNA-mediated translational repression (Fig. 1A). When a transcript is bound to AGO2, this indicates its incorporation into the RISC in which translation of the mRNA is likely regulated by miRNAs. Thus, the proportion of AGO2-bound mRNA can indicate the extent of miRNA-mediated translational control. AGO2 was immunoprecipitated from control and iLTP-treated neurons, and we used qRT-PCR to measure AGO2-bound *Gphn* (Fig. EV1A,B). Following iLTP

stimulation, we observed reduced binding of *Gphn* mRNA by AGO2 compared to sham control, suggesting some alleviation of miRNA-mediated translational suppression during iLTP. Furthermore, total *Gphn* levels were unchanged following stimulation, confirming that elevated gephyrin expression during iLTP is not likely due to increased transcription (Fig. EV1C).

To identify miRNAs that might mediate *Gphn* silencing, we cross-referenced miRNA seed sites in the *Gphn* 3'UTR (identified by TargetScan and miRDB) (Lewis et al, 2003; Wong and Wang, 2014) with next-generation sequencing (NGS) data of miRNAs exhibiting altered expression levels following a similar iLTP stimulation in hippocampal brain slices (Hu et al, 2014). This search led to miR153, which targets bases 61–67 of the *Gphn* 3'UTR (Fig. 1A) and is predicted to be downregulated in the hippocampus following iLTP-like stimulation (Hu et al, 2014). To verify its downregulation during iLTP, we quantified levels of mature miR153 from cultured hippocampal neurons harvested at multiple time-points following iLTP stimulation using qRT-PCR. This experiment showed that miR153 expression was gradually reduced during iLTP, in comparison with a control miRNA, miR15a (Fig. 1B). This result aligns well with the concurrent increase in gephyrin expression characterized during iLTP (Rajor et al, 2020).

### miR153 silences *Gphn* and controls endogenous gephyrin levels

We next wanted to assess whether miR153 was able to repress *Gphn* mRNA translation via binding to its 3'UTR. To do this, we used reporter plasmids where Firefly luciferase (Luc) was fused to the *Gphn* mRNA 3'UTR, so Luc expression functioned as a readout of *Gphn* translational activity. Luc constructs contained either the wild-type *Gphn* 3'UTR (Luc-*Gphn*[153-WT]) or a mutant 3'UTR (Luc-*Gphn*[153-Mut]), with mutations in the predicted seed site for miR153 (Fig. 1C) and were co-expressed with miR153 in HEK cells (Fig. 1D). Luc activity readings revealed that miR153 overexpression (miR153 OE) decreased translational activity of Luc-*Gphn*[153-WT] by ~45%, compared with control miRNA (miRCon OE). In contrast, translational activity of Luc-*Gphn*[153-Mut] was not impacted by miR153 OE. Notably, Luc-*Gphn*[153-WT] translational activity was significantly lower than that of Luc-*Gphn*[153-Mut] in miR153 OE cells, indicating that the miR153 seed site is required for miR153-mediated translational silencing of gephyrin.

To determine if the miR153 seed site is also important for *Gphn* translational activity in neurons during iLTP, we expressed the Luc-*Gphn*[153-WT] and Luc-*Gphn*[153-Mut] reporters in cultured hippocampal neurons and measured their translational activity (Fig. 1E). Luc-*Gphn*[153-Mut] exhibited higher translational activity than Luc-*Gphn*[153-WT] in control conditions, demonstrating that the miR153 seed site is active and its disruption is sufficient to increase *Gphn* translation in neurons. Furthermore, the translational activity of Luc-*Gphn*[153-WT] was increased following iLTP stimulation, while Luc-*Gphn*[153-Mut] activity was unaltered, suggesting elevated *Gphn* translation during iLTP is controlled by miR153. Together, these data show that miR153 interacts with its seed site in the *Gphn* 3'UTR to suppress translation in both an in vitro reduced system and in hippocampal neurons, revealing a novel mechanism for controlling gephyrin expression. Furthermore, this suppression is relieved in neurons during iLTP, consistent with decreased miR153 neuronal expression following stimulation.

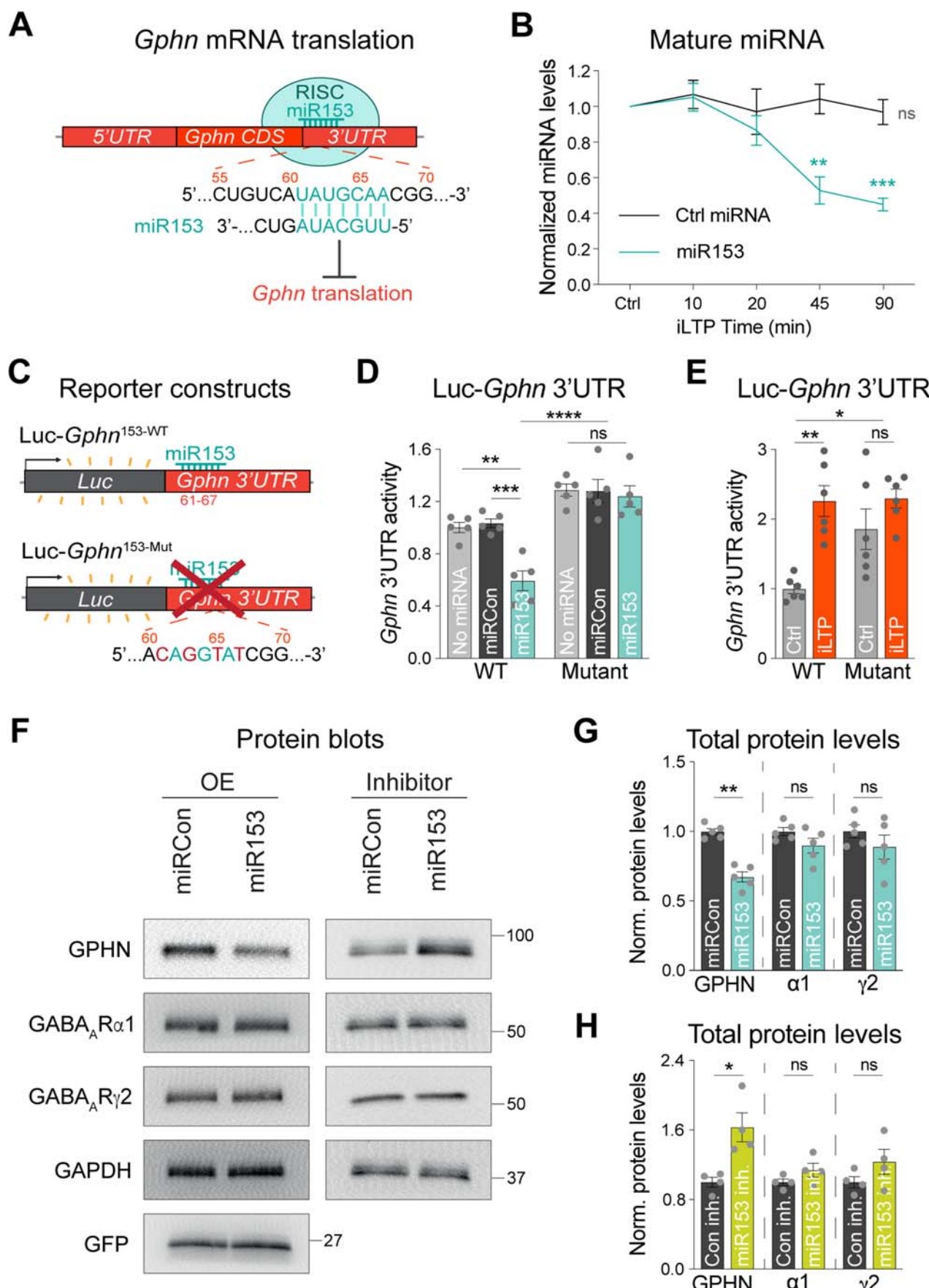

**Figure 1.  miR153 is downregulated during iLTP and controls endogenous gephyrin expression.**

(A) Schematic of RNA-Induced Silencing Complex (RISC) and miR153 interaction with the seed site in *Gphn* 3'UTR, which is predicted to suppress translation of this mRNA. (B) qRT-PCR of miR153 and miR15a (Ctrl miRNA) expression in cultured hippocampal neurons harvested at different time-points following iLTP stimulation. miRNA levels were normalized to U6. N = 4. P-values (Ctrl vs iLTP t = 10/20/45/90 min): miR15a = 0.3113/0.8460/0.6049/0.3843, miR153 = 0.3811/0.8735/0.0040/<0.0001 (C) Schematic of the Luc-*Gphn* luciferase reporters. miR153 seed site is mutated in Luc-*Gphn*[153-Mut]. (D) Quantification of Luc-*Gphn* activities in HEK293T cells co-expressing control miRNA (miRCon), miR153, or no miRNA. Firefly was normalized to Renilla, and the data quantified as relative change in normalized Luc activity. N = 5. P-values: WT no miRNA vs miR153 = 0.0005, WT miRCon vs miR153 = 0.0002, Mut no miRNA vs miR153 = 0.8692, Mut miRCon vs miR153 = 0.9018, WT miR153 vs Mut miR153 < 0.0001. (E) Quantification of Luc-*Gphn* activities in hippocampal neurons under control conditions (Ctrl) or 90 min post-iLTP stimulation. N = 6. P-values: WT Ctrl vs iLTP = 0.0009, WT Ctrl vs Mut Ctrl = 0.0249, WT Ctrl vs Mut iLTP = 0.0007, Mut Ctrl vs Mut iLTP = 0.4386. (F) Western blots of gephyrin (GPHN), GABA$_A$R subunits α1 and γ2, GAPDH, and GFP protein levels in neurons overexpressing miRCon or miR153 (left), and miRCon inhibitor or miR153 inhibitor (right). miRNA overexpression (OE) constructs contain a GFP reporter. (G) Quantification of GPHN, α1, and γ2 levels in miRCon or miR153 OE neurons. Protein levels were normalized to GAPDH, and the data quantified as relative change in normalized protein expression. N = 5. P-values (miRCon vs miR153): GPHN = 0.0079, α1 = 0.1508, γ2 = 0.3095. (H) Quantification of GPHN, α1, and γ2 in neurons expressing miRCon or miR153 inhibitors. N = 4. P-values (anti-Con vs anti-153): GPHN = 0.0286, α1 = 0.2000, γ2 = 0.2000. N = independent neuronal cultures/experiments. All values represent mean ± SEM. *p < 0.05, **p < 0.01, ***p < 0.005, ****p < 0.0001; one-sample t-test (B), two-way ANOVA with Tukey's (D) or Šidák's (E) multiple comparisons post-hoc test, and Mann–Whitney test (G, H).

Given that miR153 controls *Gphn* 3'UTR activity, we would expect it to regulate endogenous gephyrin protein expression. To test this idea, we transduced hippocampal neurons with AAVs expressing miRCon or miR153 constructs and used western blotting to measure total gephyrin protein levels (Fig. 1F,G). miR153 OE robustly decreased gephyrin protein levels, with no impact on miR376c targets, GABA$_A$R α1 and γ2. Furthermore, the expression of GABA$_A$R subunits β3 and α5 (synaptic and extrasynaptic, respectively) and the AMPA receptor subunit, GluA1, remained unaltered (Fig. EV1D,E). We also found that miR153 OE had no impact on expression of GPHN-binding proteins neuroligin-2 (NL2) and collybistin (CB), providing further evidence of the specificity of miR153 for controlling GPHN levels. To validate our miR153 overexpression approach, we also probed for the expression of another validated miR153 target, VAMP2 (Mathew et al, 2016), and observed decreased VAMP2 levels in miR153 OE neurons compared to miRCon neurons. We also tested gephyrin protein expression in hippocampal neurons expressing a miR153 inhibitor, which sequesters miR153 thereby reducing expression levels (Fig. 1F,H), and revealed that miR153 inhibition was sufficient to elevate gephyrin protein levels compared to a control inhibitor (miRCon inh.). Similarly, total levels of GABA$_A$R subunits α1, γ2, β3, and α5, as well as GluA1 and NL2 were unaffected by miR153 inhibition (Figs. 1F,H and EV1D,F), demonstrating that manipulating miR153 levels is sufficient to drive changes in endogenous gephyrin protein expression with little impact on other key synaptic proteins.

## miR153 controls gephyrin and GABA$_A$R synaptic clustering

Since manipulation of miR153 levels impacted gephyrin translation and total protein levels, we next wanted to determine whether miR153 could control gephyrin expression at synapses. We first assessed gephyrin synaptic clustering in hippocampal cultures overexpressing miR153 or miRCon (GFP reporter, expressed for 48–72 h), using immunocytochemistry (ICC) with antibodies to gephyrin and the vesicular GABA transporter (VGAT), a marker for GABAergic presynaptic terminals (Fig. 2A). For these experiments, sparse transfection of the miRNA constructs allowed us to analyze the impact of miR153 OE specifically in the post-synaptic

cell and assess cell-autonomous effects. Confocal imaging and cluster analysis revealed that miR153 OE decreased the area and density of gephyrin and VGAT clusters in neuronal dendrites (Fig. 2B), suggesting that the effects of miR153 on gephyrin expression can alter gephyrin synaptic clustering and impact the number and size of inhibitory synapses. Analysis of GPHN puncta co-localized with VGAT showed that miR153 OE had a similar effect on VGAT[+] GPHN puncta (Fig. 2C). miR153 had little effect on somatic inhibitory synapses (Fig. EV2A,B,E), suggesting potential compartment specificity of miR153 in controlling gephyrin synaptic expression.

Since gephyrin supports GABAergic synaptic structure and function (Agarwal et al, 2008; Smith and Kittler, 2010; Fang et al, 2011; González, 2013; Hales et al, 2013; Dejanovic et al, 2014; Tyagarajan and Fritschy, 2014; Kiss et al, 2016; Dejanovic et al, 2015; Lorenz-Guertin and Jacob, 2018; Mele et al, 2019), we reasoned that miR153 may also impact GABA$_A$R synaptic clusters. To test this hypothesis, we quantified surface GABA$_A$R-γ2, which serves as a readout of synaptic GABA$_A$Rs (sGABA$_A$Rs), and VGAT in neurons transfected with miR153 or miRCon OE constructs (Fig. 2D–F). miR153 OE caused decreased sGABA$_A$R and VGAT clustering in dendrites, again with no effect on GABA$_A$Rs at inhibitory synapses in the soma (Fig. EV2C,D,F). Similarly to what we observed for VGAT[+] GPHN, analysis of VGAT[+] sGABA$_A$R clusters showed that GABA$_A$Rs opposed to a GABAergic presynaptic terminal is significantly disrupted by miR153 OE. The overall co-localization of VGAT[+] GPHN or sGABA$_A$Rs was not impacted by miR153 (Fig. EV2G). Furthermore, excitatory synaptic clusters were unaltered in miR153 OE neurons, indicating that miR153 specifically affects inhibitory connections by manipulating gephyrin expression in the postsynaptic neuron (Fig. EV2H,I). We also wanted to explore how inhibiting miR153 affected GPHN clusters, since sequestration of miR153 was sufficient to increase total GPHN protein levels (Fig. 1). ICC showed no impact of miR153 inhibitor (miR153 inh.) on GPHN cluster area or density (Fig. EV2J,K). Since many proteins are required to construct a functional inhibitory synapse and miR153 expression does not alter total expression of synaptic GABA$_A$R subunits such as α1/β3/γ2 or the inhibitory synaptic adhesion molecule NL2 (Fig. 1, Fig. EV1), we predict that increased synthesis of gephyrin protein due to miR153 inhibition is necessary but not sufficient to increase inhibitory synaptic clusters.

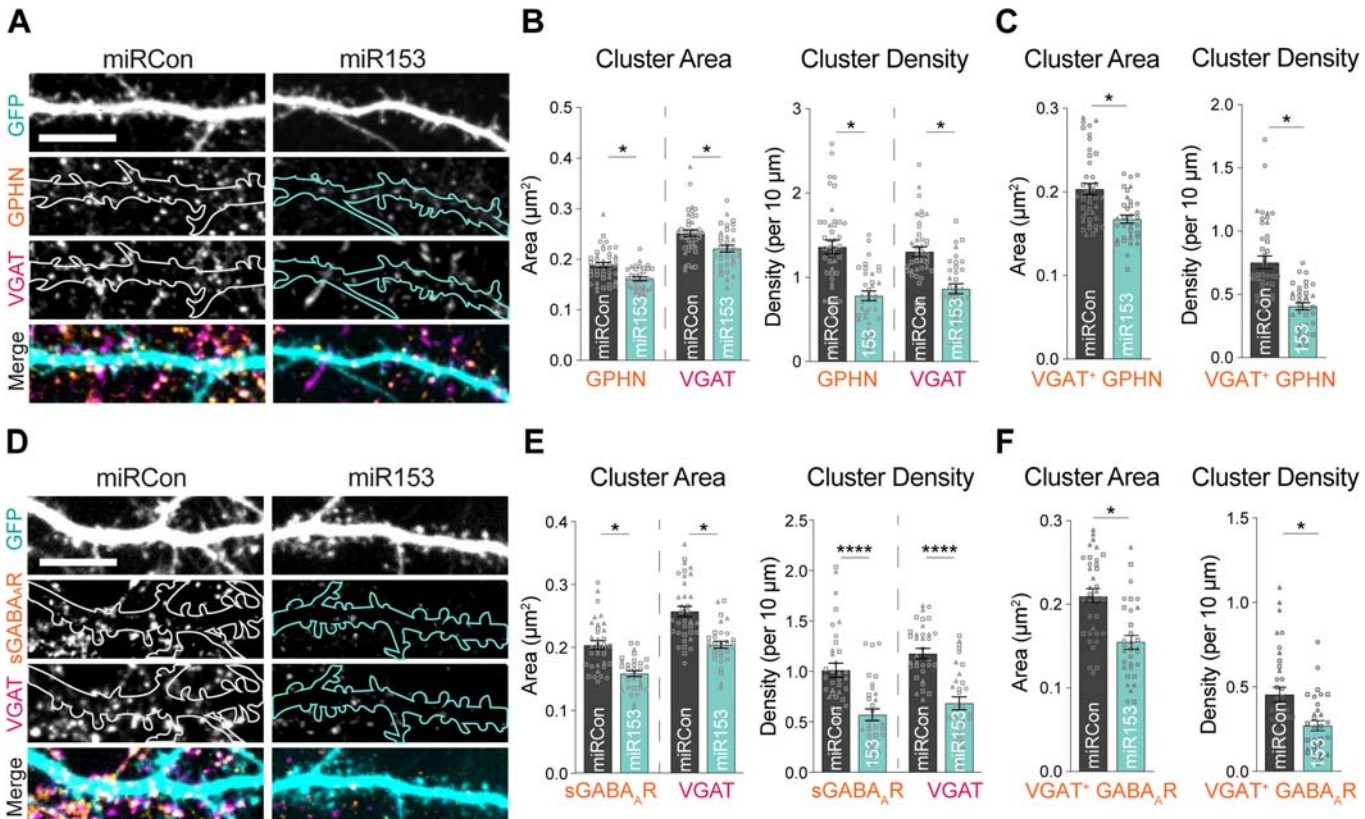

**Figure 2. miR153 overexpression disrupts gephyrin and GABA$_A$R synaptic clustering.**

(A) Representative dendritic segments of miRCon or miR153 OE-expressing neurons labeled with antibodies to gephyrin (GPHN) and VGAT. Scale bar, 10 μm. (B) Quantification of GPHN and VGAT cluster area (left) and cluster density (right) in neurons from (A). $N = 3 / n = 35$–42 neurons per condition. *P*-values (miRCon vs miR153): GPHN area = 0.0428, VGAT area = 0.0459, GPHN density = 0.0292, VGAT density = 0.0405. (C) Quantification of VGAT$^+$ GPHN cluster area (left) and density (right) from total GPHN puncta quantified in (B). $N = 3 / n = 35$–42 neurons per condition. *P*-values (miRCon vs miR153): GPHN area = 0.0143, GPHN density = 0.0495. (D) Representative dendritic segments of miRCon or miR153 OE-expressing neurons labeled with antibodies to surface GABA$_A$R subunit γ2 (sGABA$_A$R) and VGAT. Scale bar, 10 μm. (E) Quantification of sGABA$_A$R and VGAT cluster area (left) and cluster density (right) in neurons from (C). $N = 3 / n = 32$–35 neurons per condition. *P*-values (miRCon vs miR153): γ2 area = 0.0158, VGAT area = 0.0448, γ2 density <0.0001, VGAT density <0.0001. (F) Quantification of VGAT$^+$ sGABA$_A$R cluster area (left) and density (right) from total sGABA$_A$R puncta quantified in (B). $N = 3 / n = 32$–35 neurons per condition. *P*-values (miRCon vs miR153): γ2 area = 0.0406, γ2 density = 0.0412. $N$ = independent neuronal cultures/experiments, $n$ = neurons. All values represent mean ± SEM. Neurons from different culture preparations are represented by different symbols of data points. *$p < 0.05$, **$p < 0.01$, ***$p < 0.005$, ****$p < 0.0001$; nested t-test.

## miR153 impacts GABAergic synapse function

Next, we wanted to determine whether miR153-mediated disruption of gephyrin and GABA$_A$R post-synaptic clustering impacted the efficacy of inhibitory synaptic function. To address this, we used whole-cell voltage-clamp electrophysiology to measure miniature inhibitory synaptic currents (mIPSCs) in cultured neurons overexpressing miR153 or miRCon (Fig. 3A–D). miR153 OE significantly decreased mIPSC frequency, with no effect on mIPSC amplitude or kinetic properties (Figs. 3A–D and EV3A), indicating a reduction in the total number of functional inhibitory synapses. We also tested whether miR153 OE could impact inhibitory synaptic transmission in an intact circuit by injecting AAVs expressing miR153 or miRCon OE into the hippocampal CA1 region. Acute slices were prepared 2–3 weeks after injection and whole cell recordings of mIPSCs were made from GFP-expressing pyramidal neurons (Fig. 3E–H). As observed in culture, miR153 OE caused a significant decrease in mIPSC frequency compared with

miRCon and did not affect mIPSC amplitude or kinetic properties (Figs. 3F and EV3B), again suggesting fewer functional inhibitory synapses in miR153 OE neurons. The lack of impact of miR153 OE on mIPSC amplitude likely reflects the selective effect of miR153 OE on dendritic but not somatic synapses (Fig. EV2A–D), the latter of which have an augmented contribution to mIPSC measurements made at the soma due to dendritic filtering.

## iLTP induces transcriptional repression of miR153

What mechanisms lead to the downregulation of miR153 following iLTP stimulation? We previously found that iLTP stimulation drives transcriptional repression of miR376c, leading to its reduced overall expression which enables increased translation of its targets, *Gabra1* and *Gabrg2* (Rajgor et al, 2020). miR153 levels are also gradually reduced following iLTP stimulation (Fig. 1B), closely mirroring the iLTP-induced decrease in miR376c. Thus, we hypothesized that iLTP stimulation might also induce

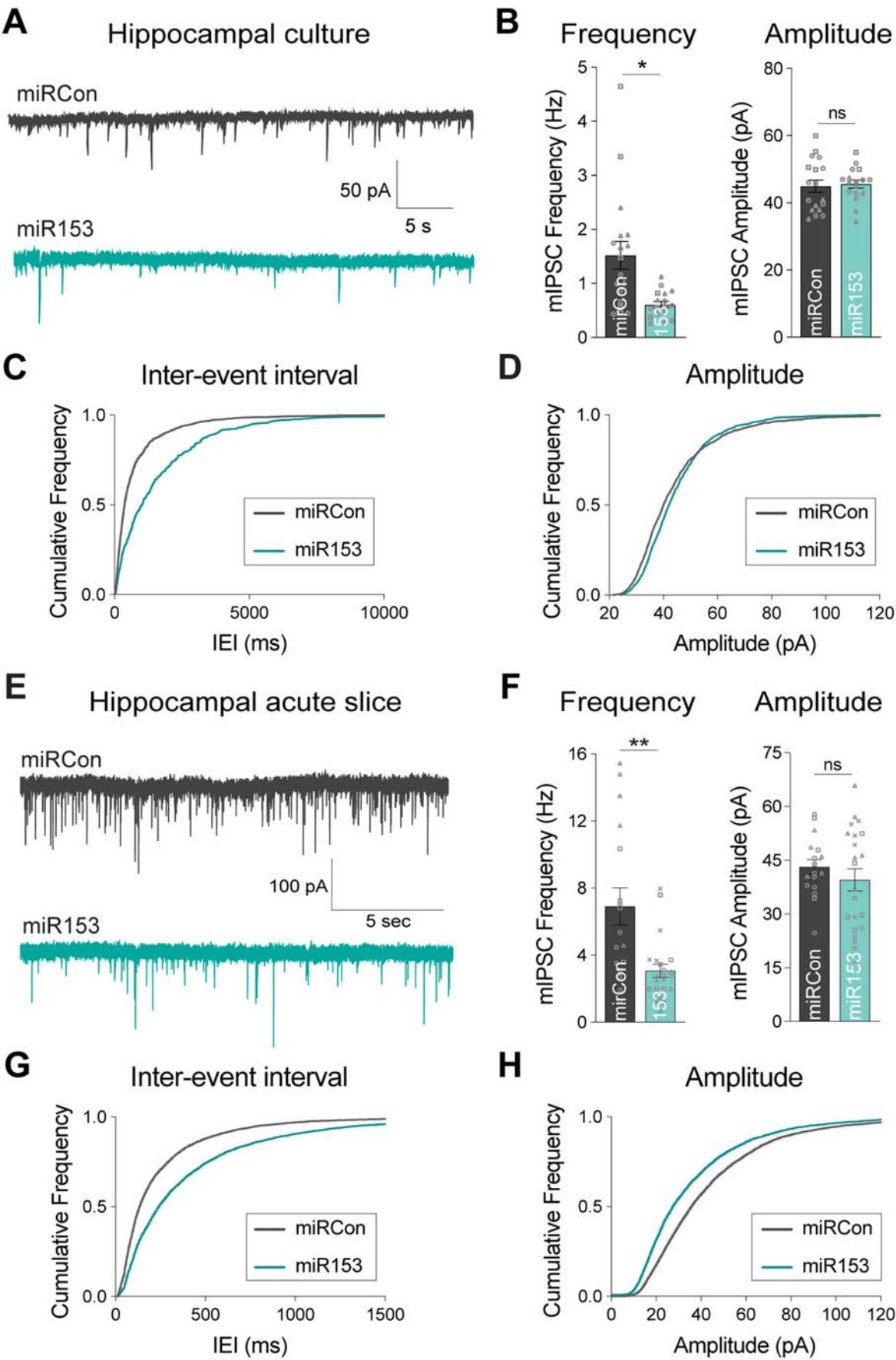

**Figure 3.** miR153 overexpression impacts GABAergic synaptic transmission.

(A) Representative mIPSC current traces from miRCon and miR153 OE-expressing neurons in hippocampal culture. (B) Quantification of mIPSC frequency (left) and amplitude (right) from miRCon and miR153 OE-expressing neurons. $N = 3 / n = 17$–18 neurons per condition. P-values (miRCon vs miR153): frequency = 0.0366, amplitude = 0.7475. (C) Cumulative frequency distribution of mIPSC inter-event intervals (IEI) for events in miRCon and miR153 OE-expressing neurons. (D) Cumulative frequency distribution of mIPSC amplitude for events in miRCon and miR153 OE-expressing neurons. (E) Representative traces recorded from miRCon and miR153 OE-expressing neurons in acute hippocampal slices. (F) Quantification of mIPSC frequency (left) and amplitude (right) from miRCon and miR153 OE-expressing neurons. $N = 3$–4/ $n = 17$–22 neurons per condition. P-values (miRCon vs miR153): frequency = 0.0041, amplitude = 0.3327. (G) Cumulative frequency distribution of mIPSC IEI for events in miRCon and miR153 OE-expressing neurons. (H) Cumulative frequency distribution of mIPSC amplitude for events in miRCon and miR153 OE-expressing neurons. $N =$ independent neuronal cultures/experiments, $n =$ neurons. All values represent mean ± SEM. Neurons from different neuronal preparations are represented by different symbols of data points. *$p < 0.05$, **$p < 0.01$, ***$p < 0.005$, ****$p < 0.0001$; nested (B) or Welch's (F) t-test.

transcriptional repression of miR153. To test this hypothesis, we used qRT-PCR to assess levels of the miR153 primary transcript (pri-miR153) following iLTP stimulation. Primary miRNAs are the initial transcripts which are processed into mature miRNAs and eventually degraded. Thus, quantification of pri-miRNAs can function as a readout of gene transcription. Pri-miR153 expression was drastically reduced within 10 min of iLTP stimulation (Fig. 4A), indicating its rapid transcriptional downregulation. To assess miR153 half-life in neurons under basal conditions, we used actinomycin-D (ActD) to inhibit transcription for up to 90 min and measured mature miR153 levels (Fig. 4B). With ActD treatment, miR153 expression rapidly decreased over time, with levels reduced by ~50% at 20 min and ~75% by 90 min. This suggests that miR153 has a relatively high turnover rate in neurons, similar to miR376c (Rajgor et al, 2020). Thus, this short half-life, in combination with its transcriptional repression, likely leads to the gradual reduction in miR153 neuronal expression following iLTP stimulation.

## iLTP-induced miR153 transcriptional repression is controlled by calcineurin

During iLTP-induced E-T coupling, NMDARs and L-type calcium channels (LTCCs) activate calcineurin (CaN) to facilitate transcriptional repression of miR376c via NFAT and HDAC activity (Rajgor et al, 2020). Since iLTP stimulation also downregulates miR153 through transcriptional repression, we reasoned that a similar E-T coupling pathway could also repress miR153 transcription, to enable coordinated upregulation of gephyrin alongside synaptic GABA$_A$R subunits. As CaN activation is critical for the downstream E-T signaling to suppress miR376c, we first tested whether iLTP-induced reduction in miR153 also requires CaN activity. We used qRT-PCR to quantify pri-miR153 levels after iLTP stimulation, in the presence or absence of CaN inhibitors cyclosporin A (CsA) or FK506 (Fig. 4C,D). Indeed, blockade of CaN activity completely prevented the reduction in pri-miR153 expression during iLTP (Fig. 4C). Moreover, CaN inhibition also prevented the downregulation of mature miR153 (Fig. 4D), indicating that the reduced expression of miR153 during iLTP is controlled by CaN activation and potentially by a similar E-T coupling pathway as miR376c.

## NFATc3 and HDACs control the concurrent transcriptional repression of miR153 and miR376c

During iLTP, transcriptional repression of miR376c is dependent on two NFAT binding sites located at −125 and −109 bp upstream of the precursor miR376c (pre-miR376c) coding region. Our data

suggest a model whereby CaN activation promotes NFATc3 translocation to the nucleus, where it facilitates HDAC-dependent epigenetic repression of miR376c transcription (Rajgor et al, 2020). As with miR376c, we identified a putative NFATc3 binding site -65 bp upstream of the pre-miR153 coding region (Fig. 4E), suggesting that NFATc3 could also modulate miR153 transcription via a similar mechanism. We first determined if the NFATc3 binding site was active by fusing luciferase to the 500 bp sequence upstream of the pre-miR153 coding region (miR153$^{NFAT-WT}$-Luc; Fig. 4E) to provide a readout of miR153 transcriptional activity. When expressed in neurons, miR153$^{NFAT-WT}$-Luc exhibited an ~5-fold increase in Luc activity compared with an empty luciferase vector, suggesting this region is transcriptionally active (Fig. EV4). Following iLTP induction, the activity of miR153$^{NFAT-WT}$-Luc was reduced by ~60% at 90 min (Fig. 4F) and this reduction was completely prevented by the inclusion of CaN inhibitors, again confirming a significant role for CaN activation in controlling miR153 transcription (Fig. 4G). Crucially, when the NFATc3 binding site in miR153 was mutated (miR153$^{NFAT-Mut}$-Luc), the iLTP-induced reduction in transcriptional activity was blocked (Fig. 4F). This result suggests that the NFATc3 binding site upstream of the pre-miR153 coding region is active and likely important in regulating miR153 transcription during iLTP.

We also wanted to show that endogenous NFATc3 could repress the transcriptional activity of these upstream regulatory regions of both miR153 and miR376c. To do this, we used shRNA to knockdown NFATc3 (NFAT KD, Fig. EV4C) and Luc assays to measure the impact of NFATc3 KD on miR376c and miR153 transcriptional activity (Fig. 4H). Luc measurements revealed that NFAT KD was sufficient to increase transcriptional activity of both miR376c$^{NFAT-WT}$-Luc and miR153$^{NFAT-WT}$-Luc in hippocampal neurons, compared to control knockdown (Ctrl KD) and NFATc3 knockdown plus rescue (KD + Rescue) (Fig. 4H). This concurrent increase in the activity of regulatory sequences for both miRNAs establishes a requirement for NFATc3 in reducing the transcription of both miR376c and miR153 during iLTP and suggests a convergent pathway regulating their expression.

Next, we wanted to determine whether HDACs were mediators of miR153 transcriptional repression, as deacetylation of the miR376c promoter region (a mark for gene silencing) is crucial for controlling its transcriptional repression and ultimately GABA$_A$R expression during iLTP (Rajgor et al, 2020). miR153 expression is regulated by histone acetylation state in other cell types, suggesting that HDACs could indeed impact the levels of miR153 (Xu et al, 2011). To test if the miR153 regulatory region is de-acetylated during iLTP, we performed chromatin immunoprecipitation (ChIP) assays to quantify the association of acetylated histone H3 within the upstream sequence for miR153 (Fig. 4I). Like miR376c, acetyl-H3 association with miR153

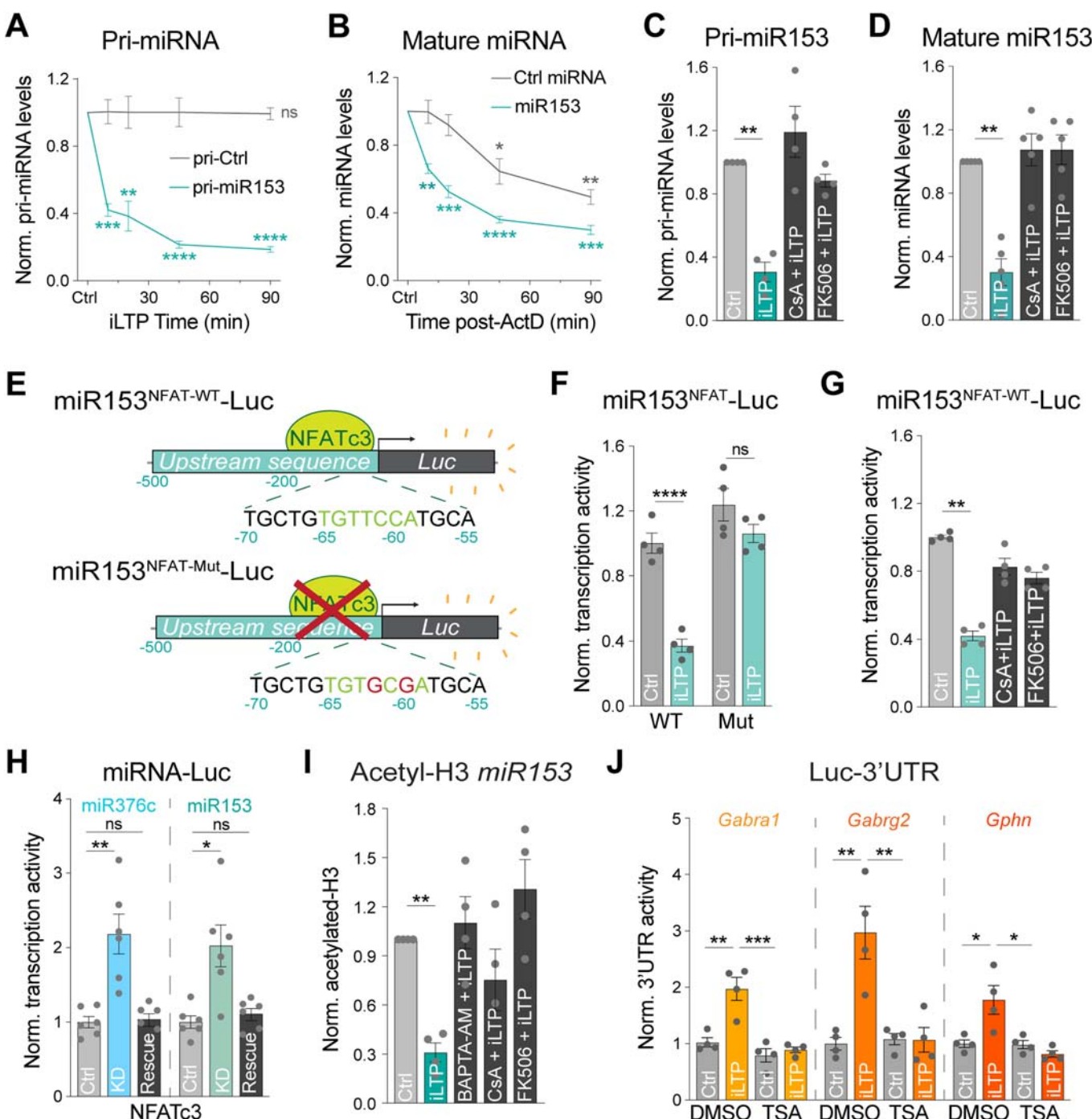

was substantially reduced following iLTP stimulation, a process that was blocked by Ca²⁺ chelation (BAPTA-AM) or CaN inhibition (CsA/FK506). This suggests that miR153 deacetylation could contribute to its transcriptional repression following iLTP stimulation and requires upstream Ca²⁺-CaN signaling. To confirm the role of HDACs in this signaling pathway, we measured the translational activity of miR376c targets *Gabra1* and *Gabrg2* as well as miR153 target, *Gphn* during iLTP, in the presence or absence of trichostatin A (TSA), which inhibits class I and II HDACs (Fig. 4J). As expected, *Gabra1*, *Gabrg2*, and *Gphn* all exhibited elevated translational activity following iLTP

stimulation. However, this was blocked by TSA treatment, demonstrating that HDACs are required for miRNA-mediated increases in inhibitory synaptic gene translation during iLTP.

## miR153 downregulation is required for increased GABAergic synaptic clustering during iLTP

Manipulation of miR153 levels alters gephyrin expression and GABAergic transmission, and miR153 levels are reduced following iLTP stimulation. Thus, we predicted that preventing miR153

◄

**Figure 4. miR153 and miR376c transcriptional repression are controlled by a common CaN-NFAT signaling pathway during iLTP.**

(A) qRT-PCR of primary miR153 transcript (pri-miR153) and pri-miR410 (pri-Ctrl) expression in neurons harvested at increasing time-points following iLTP stimulation. pri-miRNA levels were normalized to U6 and quantified as fold change from Ctrl condition. $N = 4$. P-values (Ctrl vs iLTP $t = 10/20/45/90$ min): pri-miR410 = 0.9570/0.9826/0.9874/0.8777, pri-miR153 = 0.0006/0.0063/<0.0001/<0.0001. (B) qRT-PCR of mature miR153 and miR410 (Ctrl miRNA) expression in neurons harvested at increasing time-points following treatment with actinomycin-D (ActD). miRNA levels normalized to U6 and quantified as fold change from Ctrl condition. $N = 4$. P-values (Ctrl vs ActD $t = 10/20/45/90$ min): miR410 = 0.9773/0.2783/0.0186/0.0014, miR153 = 0.0012/0.0009/<0.0001/<0.0001. (C) qRT-PCR of pri-miR153 expression in hippocampal neurons following control treatment (Ctrl) or 90 min post-iLTP stimulation in the presence or absence of CaN inhibitors cyclosporin A (CsA) and FK506. Quantified as fold change in pri-miR153 levels from Ctrl condition. $N = 4$. P-values (Ctrl vs iLTP): DMSO = 0.0015, CsA = 0.3163, FK506 = 0.0675. (D) qRT-PCR of mature miR153 expression in Ctrl and iLTP-90 neurons in the presence or absence of CsA and FK506. Quantified as miR153 fold change from Ctrl condition. $N = 5$. P-values (Ctrl vs iLTP): DMSO = 0.0012, CsA = 0.5027, FK506 = 0.4661. (E) Schematic of the miR153-Luc luciferase reporters. Predicted NFAT binding site is mutated in miR153[NFAT-Mut]-Luc. (F) Quantification of miR153-Luc activities in neurons under control conditions (Ctrl) or 90 min post-iLTP stimulation. Firefly was normalized to Renilla, and the data quantified as relative change in normalized Luc activity with error-corrected control values. $N = 4$. P-values (Ctrl vs iLTP): WT = 0.0001, Mut = 0.3094. (G) Quantification of miR153-Luc activities in Ctrl and iLTP-90 neurons in the presence or absence of CsA and FK506. Quantified as fold change in miR153-Luc activity from Ctrl condition. $N = 4$. P-values (Ctrl vs iLTP): DMSO = 0.0011, CsA = 0.4125, FK506 = 0.1128. (H) Quantification of miR376c-Luc and miR153-Luc activities in Ctrl, NFATc3 knockdown (KD) and NFATc3 KD + rescue (Rescue) neurons. $N = 6$. P-values: miR376c Ctrl vs KD = 0.0059, miR376c Ctrl vs Rescue >0.9999, miR153 Ctrl vs KD = 0.0116, miR153 Ctrl vs Rescue >0.9999. (I) qPCR readout of acetyl-histone H3 chromatin immunoprecipitation (ChIP) from neurons to show acetylation status of the miR153 promoter in Ctrl and iLTP-90 conditions in the presence or absence BAPTA-AM, CsA, and FK506. $N = 4$. P-values (Ctrl vs iLTP): DMSO = 0.0013, BAPTA-AM = 0.5615, CsA = 0.2802, FK506 = 0.1865. (J) Quantification of Luc-Gabra1, Luc-Gabrg2, and Luc-Gphn activities in Ctrl and iLTP-90 neurons in the presence or absence of HDAC inhibitor trichostatin-A (TSA). $N = 4$. P-values: *Gabra1* DMSO Ctrl vs iLTP = 0.0010, *Gabra1* DMSO iLTP vs TSA Ctrl = 0.0001, *Gabra1* DMSO iLTP vs TSA iLTP = 0.0003; *Gabrg2* DMSO Ctrl vs iLTP = 0.0011, *Gabrg2* DMSO iLTP vs TSA Ctrl = 0.0016, *Gabrg2* DMSO iLTP vs TSA iLTP = 0.0015; *Gphn* DMSO Ctrl vs iLTP = 0.0092, *Gphn* DMSO iLTP vs TSA Ctrl = 0.0081, *Gphn* DMSO iLTP vs TSA iLTP = 0.0019. $N$ = independent neuronal cultures/experiments. All values represent mean ± SEM. *$p < 0.05$, **$p < 0.01$, ***$p < 0.005$, ****$p < 0.0001$; one-sample t-test (A–D, I), two-way ANOVA with Šidák's (F) or Tukey's (J) multiple comparisons post-hoc test, and Kruskal–Wallis with Dunn's multiple comparisons post-hoc test (G, H).

downregulation during iLTP via miR153 OE would disrupt plasticity-induced changes at inhibitory synapses. To test this prediction, we developed a live-imaging assay to track inhibitory synapse growth and formation in the same cell over time, using the gephyrin intrabody (GPHN-IB) to label endogenous gephyrin (Gross et al, 2013; Crosby et al, 2019), and live antibody labeling of VGAT with VGAT-Oyster[650]. As we have previously observed in fixed imaging experiments (Rajgor et al, 2020), live imaging revealed a steady increase in gephyrin and VGAT cluster intensity and density in the 90 min following iLTP stimulation, showing an increase in the number and size of inhibitory synapses during iLTP (Fig. EV5A–C). Importantly, application of the translational inhibitor cycloheximide (CHX), blocked the maintenance of this increased clustering at 90 min post-stimulation, recapitulating our findings using fixed-cell confocal imaging (Rajgor et al, 2020). We then used this approach to determine how preventing miR153 downregulation would impact iLTP-induced increases in inhibitory size and density (Figs. 5A–C and EV5D). We sparsely expressed GPHN-IB and fluorescently tagged miRCon or miR153 OE constructs (~72 h), live labeled with VGAT-Oyster[650], and imaged neuronal dendrites in sham or iLTP conditions. As expected, gephyrin synaptic cluster intensity increased steadily in miRCon OE neurons treated with iLTP conditions compared with sham-stimulated controls. In contrast, miR153 OE-expressing neurons exhibited no increase in inhibitory synaptic clustering over time following stimulation, suggesting that preventing reduced miR153 expression during iLTP stimulation is sufficient to disrupt persistent iLTP-dependent increases in synaptic clustering and density.

As miR153 OE prevented increased synaptic clustering of gephyrin and VGAT following iLTP stimulation, we then examined whether this also impacts the iLTP-induced upregulation of GABA_AR clusters at synapses (Fig. 5D,E). ICC experiments using neurons expressing miRCon or miR153 and confocal imaging revealed an expected increase in GABA_AR and VGAT clusters in miRCon OE neurons at 90 min following iLTP, while this elevated synaptic clustering of GABA_ARs/VGAT was blocked by miR153 OE. These data indicate that the reduction in miR153 expression following iLTP stimulation is a crucial mechanism regulating GABAergic synapse clustering during persistent iLTP.

## Inhibitory synaptic potentiation during iLTP requires reduced miR153 expression

Our data have demonstrated that overexpressing miR153 decreased basal inhibitory synaptic function (Fig. 3) and that preventing reduced miR153 expression following stimulation prevents the increase of GPHN/GABA_AR synaptic clustering during iLTP (Fig. 5). To assess the functional impact of miR153 OE on iLTP, we repeated whole-cell voltage-clamp recordings of mIPSCs in cultured hippocampal neurons expressing miR153 OE or miRCon in both basal and iLTP conditions. In miRCon-expressing neurons, mIPSC frequency increased during iLTP compared to sham treatment. Furthermore, miR153 OE neurons showed reduced mIPSC frequency in both sham and iLTP conditions as compared to control miRCon neurons (Fig. 6A–C). These results recapitulated what we previously observed during basal activity (Fig. 3), and showed that reduced miR153 expression is required for synaptic potentiation following iLTP stimulation.

## CaN signaling orchestrates sustained inhibitory synapse upregulation during iLTP

Our data show that following iLTP stimulation, CaN signaling controls downstream transcriptional repression of both miR376c and miR153 (Rajgor et al, 2020). However, it is unclear if this signaling ultimately leads to sustained increases in inhibitory synapse size and density during iLTP. To first assess whether CaN activity is required for *Gphn* translation, we performed Luc reporter assays to measure the translational activity of the *Gphn* 3'UTR during iLTP (Fig. 7A). Inclusion of BAPTA or CsA/FK506 prevented the iLTP-induced increase in translational activity of

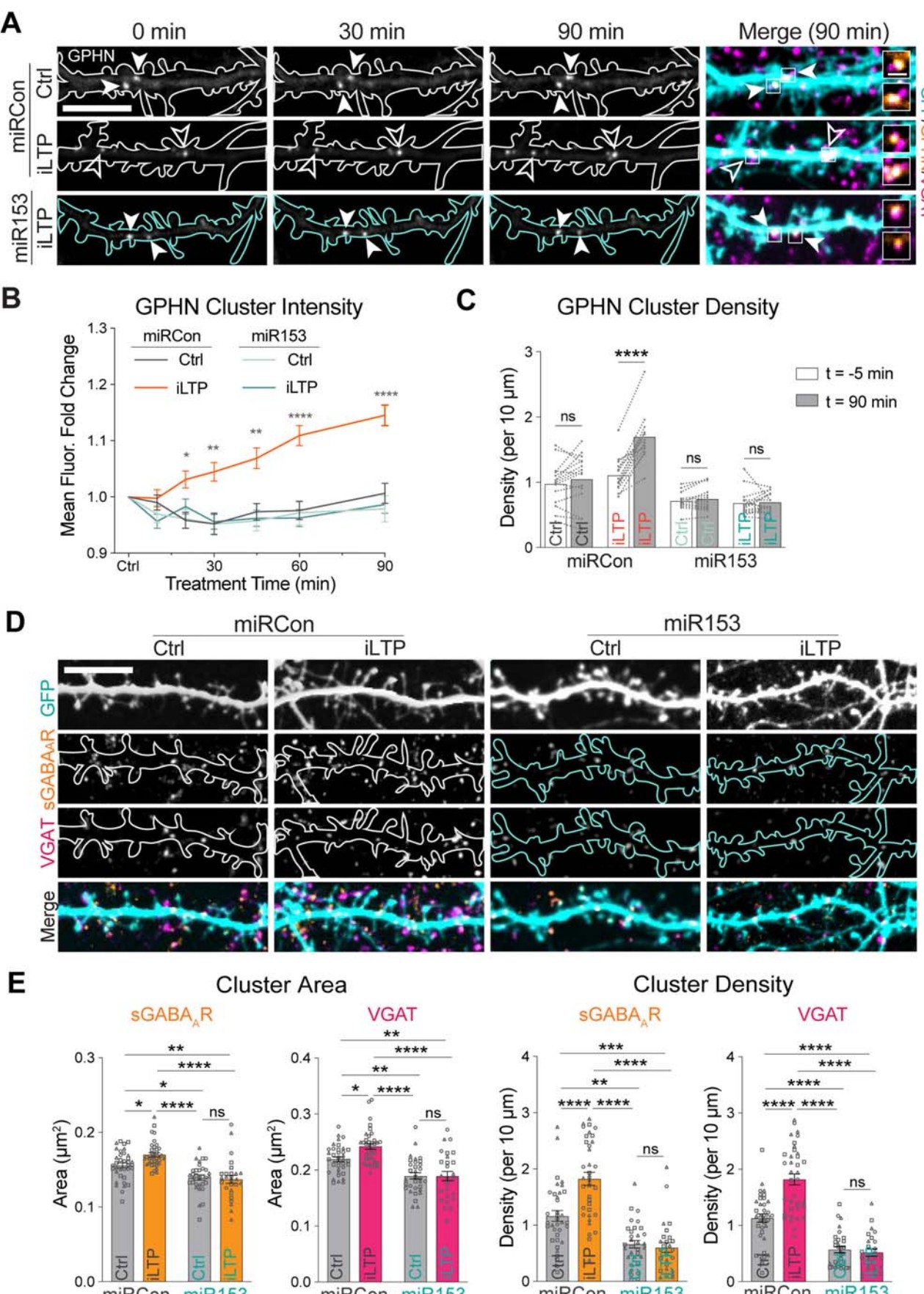

◀ **Figure 5.  miR153 overexpression prevents increased GABAergic synaptic clustering during iLTP.**

(A) Representative dendritic segments of miRCon or miR153 OE-expressing neurons over time in control and iLTP conditions. Neurons co-expressed the gephyrin intrabody (GPHN IB, arrowheads) and labeled live with VGAT Oyster[650]. Puncta are labeled with filled arrowheads when the fluorescence is unchanged and open arrowheads when fluorescence increases over time. Boxes indicate the fluorescent puncta enlarged in the merged images (dendrite scale bar, 10 µm; synapse scale bar, 2 µm). (B) Quantification of fold change in GPHN puncta fluorescence intensity over time following treatment in neurons from (A). $N = 3 / n = 15$ neurons per condition. *P*-values (miRCon Ctrl vs iLTP): 10 min > 0.9999, 20 min = 0.0125, 30 min = 0.0049, 45 min = 0.0012, 60 min < 0.0001, 90 min < 0.0001. (C) Paired measurements of GPHN cluster density in dendrites prior to (−5 min) and 90 min following treatment. $N = 3 / n = 15$ neurons per condition. *P*-values ($t = -5$ min vs $t = 90$ min): GPHN density miRCon Ctrl = 0.9137, GPHN density miRCon iLTP < 0.0001, GPHN density miR153 Ctrl = 0.9965, GPHN density miR153 iLTP = 0.9999. (D) Representative dendritic segments of miRCon or miR153 OE-expressing neurons labeled with antibodies to surface GABA$_A$R γ2 subunit (sGABA$_A$R) and VGAT following control treatment or 90 min post-iLTP stimulation. Scale bar, 10 µm. (E) Quantification of sGABA$_A$R and VGAT cluster area (left) and density (right) in neurons from (D). $N = 3/n = 27$–36 neurons per condition. P-values: γ2 area miRCon Ctrl vs iLTP = 0.0270, γ2 area miRCon Ctrl vs miR153 Ctrl = 0.0110, γ2 area miRCon Ctrl vs miR153 iLTP = 0.0040, γ2 area miRCon iLTP vs miR153 Ctrl < 0.0001, γ2 area miRCon iLTP vs miR153 iLTP < 0.0001, γ2 area miR153 Ctrl vs iLTP = 0.9985; VGAT area miRCon Ctrl vs iLTP = 0.0179, VGAT area miRCon Ctrl vs miR153 Ctrl = 0.0016, VGAT area miRCon Ctrl vs miR153 iLTP = 0.0024, VGAT area miRCon iLTP vs miR153 Ctrl < 0.0001, VGAT area miRCon iLTP vs miR153 iLTP < 0.0001, VGAT area miR153 Ctrl vs iLTP > 0.9999; γ2 density miRCon Ctrl vs iLTP < 0.0001, γ2 density miRCon Ctrl vs miR153 Ctrl = 0.0016, γ2 density miRCon Ctrl vs miR153 iLTP = 0.0006, γ2 density miRCon iLTP vs miR153 Ctrl < 0.0001, γ2 density miRCon iLTP vs miR153 iLTP < 0.0001, γ2 density miR153 Ctrl vs iLTP = 0.9989; VGAT density miRCon Ctrl vs iLTP < 0.0001, VGAT density miRCon Ctrl vs miR153 Ctrl < 0.0001, VGAT density miRCon Ctrl vs miR153 iLTP < 0.0001, VGAT density miRCon iLTP vs miR153 Ctrl < 0.0001, VGAT density miRCon iLTP vs miR153 iLTP < 0.0001, VGAT density miR153 Ctrl vs iLTP = 0.9986. $N$ = independent neuronal cultures/experiments, $n$ = neurons. All values represent mean ± SEM. *$p < 0.05$, **$p < 0.01$, ***$p < 0.005$, ****$p < 0.0001$; mixed-effects model with Geisser-Greenhouse correction (B) and Šidák's multiple comparisons post-hoc test (B, C) and ordinary two-way ANOVA with Tukey's multiple comparisons post-hoc test (E).

*Gphn* 3'UTR, implicating Ca$^{2+}$-CaN activity in the pathway driving gephyrin translation during iLTP. Furthermore, analysis of total gephyrin levels following stimulation revealed that Ca$^{2+}$-chelation or CaN inhibition could also robustly block the increase in gephyrin protein expression during iLTP (Fig. 7B,C). Given that CaN mediates the upregulation of GABA$_A$R translation and protein expression during iLTP (Rajgor et al, 2020), we then wanted to assess whether CaN acts as the primary signal for the persistent increase in inhibitory synapse size and number following iLTP stimulation. Again, we performed live-imaging of gephyrin and VGAT clusters over time following iLTP stimulation. Inhibition of CaN during iLTP blocked increased synaptic clustering of gephyrin, and the growth and formation of GABAergic synapses compared with control conditions (Figs. 7D–F and EV5F). These results mirrored what we observed in miR153 OE neurons (Fig. 5), demonstrating that preventing reduced miR153 expression during iLTP through its overexpression or via disruption of Ca$^{2+}$-CaN signaling is sufficient to block elevated inhibitory synaptic upregulation during iLTP. Altogether, these results characterize key players for altering miR153 expression following iLTP stimulation and establish a shared signaling pathway which leverages miR376c and miR153 to control changes in gene expression for multiple transcripts during iLTP.

## Discussion

In neurons, proteins are continuously synthesized and degraded to shape the synaptic proteome (Sutton and Schuman, 2006; Cajigas et al, 2010). miRNAs are crucial regulators of synaptic protein production, and therefore can significantly impact the protein composition at synapses, modifying synaptic strength and controlling various types of synaptic plasticity (Soutschek and Schratt, 2023). In previous work, we showed that gephyrin and GABA$_A$R expression at synapses increased and was maintained over time following iLTP stimulation, a process dependent on translation (Rajgor et al, 2020). Although we found that miR376c-controlled synaptic GABA$_A$R subunit translation during this process, it still remained unclear what

post-transcriptional mechanisms regulated the upregulation of the crucial scaffold, gephyrin, during iLTP. Here, we now show that a different miRNA, miR153, controls the synthesis of gephyrin following iLTP stimulation. This finding reveals a complementary mechanism to upregulate gephyrin, alongside miR376c-controlled translation of synaptic GABA$_A$Rs, and hints that numerous concurrent post-transcriptional mechanisms may coordinate de novo synthesis of the myriad proteins required to strengthen synaptic inhibition during plasticity.

## miRNA-dependent post-transcriptional regulation of gephyrin translation

Given the central role of gephyrin in mediating GABAergic synaptic transmission (Agarwal et al, 2008; Smith and Kittler, 2010; Fang et al, 2011; González, 2013; Hales et al, 2013; Dejanovic et al, 2014; Tyagarajan and Fritschy, 2014; Kiss et al, 2016; Dejanovic et al, 2015; Mele et al, 2019), it is surprising how little is known about mechanisms controlling its translation. Here, we discovered a mechanism whereby miR153 represses the translation of gephyrin in neurons under basal conditions; this repression is relieved following iLTP stimulation, allowing for gephyrin de novo synthesis. It is highly likely that miR153 is not the only factor which regulates gephyrin expression at the post-transcriptional level. Screening of the rodent *Gphn* 3'UTR predicted seed sites for eight additional miRNAs, indicating that other non-coding RNAs may also control gephyrin synthesis. In addition, multiple RNA-binding proteins (RBPs), including Nova, Staufen2, Rbfox1-3, and Pumilio2 (Pum2) can target *Gphn* mRNA (Ule et al, 2003; Lee et al, 2016; Sharangdhar et al, 2017; Zahr et al, 2018; Schieweck and Kiebler, 2019), indicating a broad range of potential mechanisms to control gephyrin translation. Recent work confirmed that gephyrin is a Pum2 target (Zahr et al, 2018), and identified Pum2 as a potential regulator of post-transcriptional gephyrin expression in the cerebral cortex (Schieweck et al, 2021). Further, *Gphn* mRNA can undergo extensive splicing (Prior et al, 1992; Rees et al, 2003; Paarmann et al, 2006), which recently was shown to govern its postsynaptic clustering properties at different types of inhibitory synapses and contributing to synaptic diversity (Bedet et al, 2006;

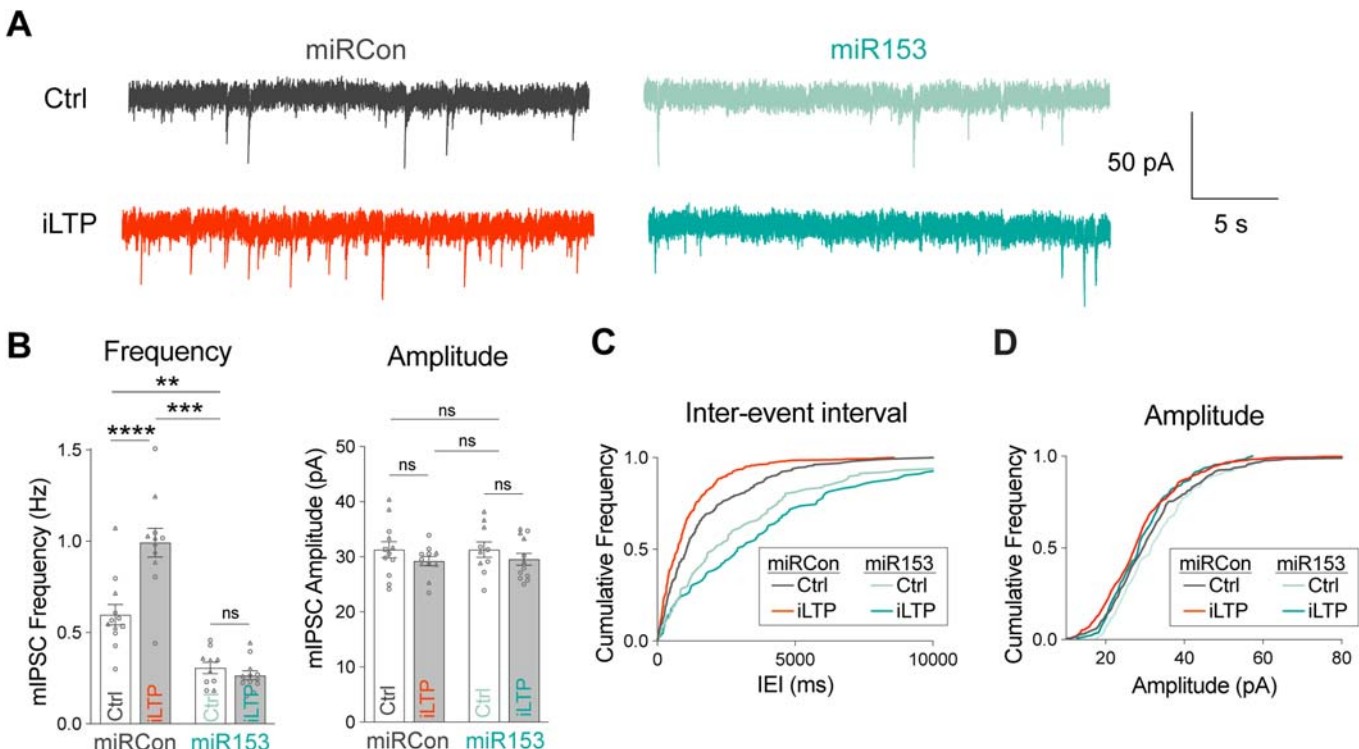

**Figure 6. miR153 overexpression prevents inhibitory synaptic potentiation following iLTP stimulation.**

(A) Representative mIPSC current traces from miRCon and miR153 OE cultured neurons in Ctrl or iLTP conditions. (B) Quantification of mIPSC frequency (left) and amplitude (right) from miRCon and miR153 OE-expressing neurons. $N = 2$ / $n = 10$–12 neurons per condition. *P*-values: frequency miRCon Ctrl vs iLTP < 0.0001, miRCon Ctrl vs miR153 Ctrl = 0.0016, miRCon Ctrl vs miR153 iLTP = 0.0001, miR153 Ctrl vs iLTP = 0.9703; amplitude miRCon Ctrl vs iLTP = 0.6500, miRCon Ctrl vs miR153 Ctrl > 0.9999, miRCon Ctrl vs miR153 iLTP = 0.7670, miR153 Ctrl vs iLTP = 0.7880. (C) Cumulative frequency distribution of mIPSC inter-event intervals (IEI) for events in miRCon and miR153 OE neurons in Ctrl or iLTP conditions. (D) Cumulative frequency distribution of mIPSC amplitude for events in miRCon and miR153 OE neurons in Ctrl or iLTP conditions. $N$ = independent neuronal cultures/experiments, $n$ = neurons. All values represent mean ± SEM. Neurons from different culture preparations are represented by different symbols of data points. *$p < 0.05$, **$p < 0.01$, ***$p < 0.005$, ****$p < 0.0001$; two-way ANOVA with Šidák's multiple comparisons post-hoc test (B).

Saiyed et al, 2007; Smolinsky et al, 2008; Tyagarajan and Fritschy, 2014; Dos Reis et al, 2022). It is likely that many of these mechanisms are brain region or cell-type specific, and possibly active during specific times in development or in different types of plasticity, thereby enabling precise control of gephyrin expression in diverse scenarios.

Overexpression of miR153 substantially reduced gephyrin and GABA$_A$R synaptic clustering in dendrites, which correlated with reduced synaptic inhibition observed in our electrophysiology experiments. Analysis revealed a substantially reduced mIPSC frequency, but no change in mIPSC amplitude, both in hippocampal culture and CA1 acute slices. This result indicates a loss of functional synapses, which could be caused by the reduction of gephyrin and GABA$_A$R synaptic clustering, if their expression decreased such that receptor activation by a single vesicle falls below the detection limit of our measurements. Furthermore, our imaging results suggest that somatic GABAergic synapses are not impacted by miR153 overexpression. Thus, changes in the amplitude of mIPSC events originating in the dendrites will likely be masked by the unaltered mIPSCs recorded in the soma, which have an outsized contribution to mIPSC measurements due to dendritic filtering of synaptic responses occurring remotely in dendrites (Papadopoulos et al, 2007; Rajgor et al, 2020). We also observed an increase in mIPSC frequency following

iLTP stimulation, which was prevented by miR153 overexpression. We did not observe a change in mIPSC amplitude during iLTP which mirrors what we have previously observed (Rajgor et al, 2020). Given that iLTP exclusively potentiates dendritic inhibitory synapses while somatic synapses remain unchanged (Chiu et al, 2018; Rajgor et al, 2020), we predict that filtering of dendritic responses recorded in the soma may explain why we do not observe increased mIPSC amplitude during iLTP, despite increased GPHN/GABA$_A$R cluster area at dendritic inhibitory synapses.

The specific mechanism which enables miR153 to exclusively impact dendritic GABAergic synaptic clustering remains unclear. Some miRNAs are enriched in neuronal dendrites and can even undergo local processing into mature miRNA transcripts in response to activity, enabling rapid, compartment-specific alterations in expression of their targets during plasticity (Schratt et al, 2006; Sambandan et al, 2017). An alternative mechanism may rely on subcellular localization of the target transcript, driven by multiple 3'UTR variants which are differentially expressed in distinct neuronal compartments (Tushev et al, 2018; Rajgor et al, 2020). However, *Gphn* mRNA does not appear to have variants with alternative 3'UTRs, suggesting that the compartment-specific effect of miR153 expression on gephyrin and GABA$_A$Rs is likely

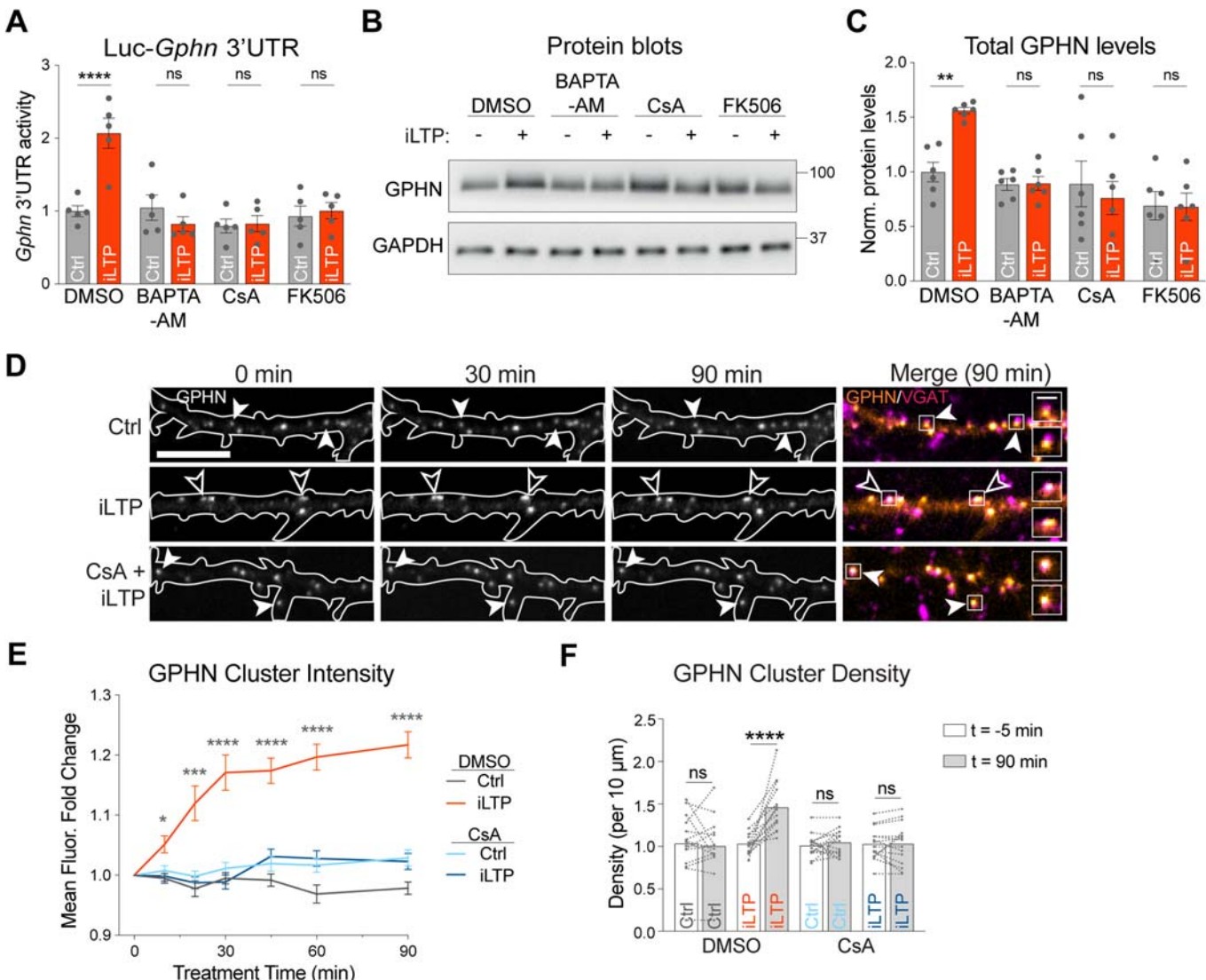

**Figure 7. Calcium and calcineurin signaling are required for increased gephyrin translation and synaptic clustering during iLTP.**

(A) Graph of Luc-*Gphn* activities in Ctrl or iLTP-90 neurons in the presence or absence of BAPTA, CsA, and FK506. Firefly was normalized to Renilla, and the data quantified as relative change in normalized Luc activity. $N = 5$. P-values (Ctrl vs iLTP): DMSO = 0.0061, BAPTA-AM > 0.9999, CsA = 0.9428, FK506 > 0.9999. (B) Western blots of GPHN and GAPDH protein levels in Ctrl and iLTP-90 neurons in the presence or absence of BAPTA, CsA, and FK506. (C) Quantification of GPHN from blots in (B). Protein levels were normalized to GAPDH, and the data quantified as relative change in normalized protein expression. $N = 6$. P-values (Ctrl vs iLTP): DMSO = 0.0074, BAPTA-AM > 0.9999, CsA = 0.9087, FK506 > 0.9999. (D) Representative dendritic segments of neurons expressing GPHN IB and labeled with an antibody to VGAT, imaged over time in control and iLTP conditions in the presence or absence of CsA. Puncta are labeled with filled arrowheads when the fluorescence is unchanged and open arrowheads when fluorescence increases over time. Boxes indicate the fluorescent puncta enlarged in the merged images (dendrite scale bar, 10 μm; synapse scale bar, 2 μm). (E) Quantification of fold change in GPHN puncta fluorescence intensity in neurons over time following treatment, as shown in (D). $N = 3$ / $n = 15$ neurons per condition. P-values (DMSO Ctrl vs iLTP): 10 min = 0.0158, 20 min = 0.0003, 30 min < 0.0001, 45 min < 0.0001, 60 min < 0.0001, 90 min < 0.0001. (F) Paired measurements of GPHN cluster density in dendrites prior to (−5 min) and 90 min following treatment. $N = 3$ / $n = 15$ neurons per condition. P-values ($t = -5$ min vs $t = 90$ min): GPHN density DMSO Ctrl = 0.9740, GPHN density DMSO iLTP < 0.0001, GPHN density CHX Ctrl = 0.9558, GPHN density CHX iLTP > 0.9999. $N$ = independent neuronal cultures/experiments, $n$ = neurons. All values represent mean ± SEM. *$p < 0.05$, **$p < 0.01$, ***$p < 0.005$, ****$p < 0.0001$; two-way ANOVA (A, C) or mixed-effects model with Geisser-Greenhouse correction (E) and Šidák's multiple comparisons post-hoc test (A, C, E, F).

due to an alternative mechanism, such as coordination with specifically-localized RBPs or the localization of miR153 itself. Given that miR153 expression selectively affects dendritic inhibitory synapses, it is tempting to speculate that local translation of gephyrin in dendrites supports changes in expression during iLTP, as we observed with GABA$_A$R mRNAs (Rajgor et al, 2020), and has

been shown for many excitatory synaptic transcripts during plasticity (Tushev et al, 2018; Biever et al, 2019; Biever et al, 2020; Donlin-Asp et al, 2021; Rajgor et al, 2021). Future work will be essential to identify the mechanism underlying the compartment-specific impact of miR153 and whether dendritic gephyrin translation is involved.

## Impact of miR153 on cellular processes and neural function

Like most miRNAs, miR153 is predicted to target numerous genes besides *Gphn*. Notably, miR153 has been shown to suppress translation of mRNAs encoding proteins involved in presynaptic active zone scaffolding and vesicle trafficking and release, including VAMP2 and SNAP25. As a result, it is possible that overexpression of miR153 may impact neurotransmitter release. For this reason, when testing how miR153 OE affects GABAergic synapse structure, function, and plasticity, we used sparse transfection of the hippocampal cultures. This approach allowed us to examine in vitro the cell-autonomous impact of miR153 OE on the postsynaptic neuron without interfering with presynaptic neurotransmitter release. As we used viral infection without cell-specific promoters to manipulate miR153 expression in slice, we cannot entirely rule out the possibility of a presynaptic effect impacting our ex vivo recordings. However, it is promising that we observe a similar impact on mIPSCs when using viral infection or sparse transfection to manipulate miR153 expression in slice and in culture, respectively. Because of its impact on presynaptic function and plasticity, miR153 is implicated in broad-reaching functions like contextual fear learning, neuronal precursor proliferation and neurogenesis, and hippocampal-dependent cognitive tasks (Mathew et al, 2016; You et al, 2019; Qiao et al, 2020; Yan et al, 2020). Furthermore, dysregulated miR153 expression has been implicated in Alzheimer's disease and autism spectrum disorders (Gupta et al, 2017; You et al, 2019; Qiao et al, 2020; Yan et al, 2020), both of which are linked to disrupted synaptic inhibition (Yizhar et al, 2011; Gao and Penzes, 2015; Busche and Konnerth, 2016; Ferguson and Gao, 2018; Selten et al, 2018; Bi et al, 2020). Our findings suggest that the role of miR153 in controlling inhibitory synaptic function and plasticity may contribute to these pathologies.

## E-T coupling coordinates transcriptional repression of miRNAs during iLTP

Our results show that miR153 expression is downregulated following iLTP stimulation via an E-T coupling pathway involving CaN, NFAT, and HDACs. This pathway mirrors the signaling that is required to downregulate miR376c during iLTP and suggests that the same signaling components control the expression of both miR153 and miR376c. Like miR376c, iLTP-induced suppression of miR153 levels is likely mediated by rapid transcriptional repression, and the subsequent degradation of the mature miRNA. This is supported by our ActD experiments which show that upon transcriptional inhibition, miR153 expression levels are rapidly reduced, indicating that miR153 has a short half-life, which is common in neural tissues like the brain and retina (Sethi and Lukiw, 2009; Krol et al, 2010; Rüegger and Großhans, 2012). However, given the time-frame of such a pronounced reduction in total miR153 levels, we cannot rule out the possibility of additional mechanisms driving decreased miR153 expression, such as activity-dependent miRNA degradation observed in neural tissue (Krol et al, 2010).

Put together with our previous work (Rajgor et al, 2020), our present findings suggest a model whereby iLTP stimulation activates CaN, which can drive the rapid translocation of NFATc3 into the nucleus. NFAT binds to regulatory sites upstream of the pre-miRNA coding sequences of miR153 and miR376c and promotes transcriptional repression through the likely recruitment of HDACs. Although the canonical promoter sequence for miR153 is located further upstream from this region (Mathew et al, 2016), we hypothesize that the sequence we used in our experiments functions as an alternative promoter or, perhaps more likely, a crucial regulatory region to modulate miR153 expression. Indeed, our data suggest that NFATc3 may recruit HDACs to alter the acetylation state of these upstream regulatory regions thus coordinately downregulating miR153 and miR376c transcription. This non-canonical role of NFATc3 in repressing miRNA transcription through an epigenetic mechanism is poorly characterized and will require further study to assess the extent of its impact in synaptic plasticity.

We have found that a single NMDA stimulation leads to dynamic alterations in the expression of two miRNAs, which target different synaptic genes in the same functional pathway. This coordinated control of miRNAs is also observed for modulation of glutamatergic synapse strength during synaptic plasticity. For instance, during excitatory LTD, miR135 and miR191 control expression of their targets (complexins and Tmod2) to coordinate the shrinkage of dendritic spines (Hu et al, 2014). Conversely, following excitatory LTP stimulation, miR26a and miR384-5p work in concert to control the maintenance of spine enlargement, a hallmark for LTP (Gu et al, 2015). Given that there are many other proteins which support inhibitory synaptic function beyond gephyrin and GABA$_A$Rs, we suspect that iLTP stimulation likely drives changes in expression for other genes involved in synaptic inhibition. It is possible that the signaling pathway we have identified here leverages an even broader miRNA network to control expression of these transcripts. Future work is aimed at identifying other genes whose translation is important for inhibitory synaptic plasticity, and whether this translation is controlled by a similar pathway. Characterizing these mechanisms is crucial to our understanding of how neurons control synaptic inhibition to maintain E/I balance and identifying potential therapeutic targets when that balance is disrupted.

# Methods

### Reagents and tools table

| Reagent/Resource | Reference or Source | Identifier or Catalog Number |
| --- | --- | --- |
| **Experimental models** | | |
| Rat, Sprague-Dawley Charles River | Charles Rivers | RRID: RGD_734476 |
| Mouse, C57BL/6 | CU Anschutz | n/a |
| Human cell line, HEK293T | ATCC | RRID:CVCL_0045 |
| **Recombinant DNA** | | |
| Gephyrin-FingR-GFP | Gift, Dr. Donald Arnold, USC (Gross et al, 2013) | RRID:Addgene_46296 |
| Gephyrin-FingR-mScarlet | Gift, Dr. Ulli Bayer, CU Anschutz | n/a |
| miRCon-GFP | Gift, Dr. Danesh Moazed, Harvard University (Mathew et al, 2016) | n/a |

| Reagent/Resource | Reference or Source | Identifier or Catalog Number |
|---|---|---|
| miR153-GFP | Gift, Dr. Danesh Moazed, Harvard University (Mathew et al, 2016) | n/a |
| pAAV-miRCon-GFP | VectorBuilder | https://en.vectorbuilder.com/design.html |
| pAAV-miR153-GFP | VectorBuilder | https://en.vectorbuilder.com/design.html |
| pMIR-REPORT-Luc-*Gphn* | This paper | n/a |
| pMIR-REPORT-Luc-*Gabra1* | Rajgor et al, 2020 | n/a |
| pMIR-REPORT-Luc-*Gabrg2* | Rajgor et al, 2020 | n/a |
| pGL4.10-miR153$^{NFAT-WT}$-Luc | This paper | n/a |
| **Antibodies** | | |
| Anti-guinea pig AlexaFluor-568 (goat) | Life Technologies | Cat. #A-11075; RRID: AB_3095468 |
| Anti-mouse AlexaFluor-568 (goat) | Life Technologies | Cat. #A-11004; RRID: AB_2534072 |
| Anti-rabbit AlexaFluor-647 (goat) | Life Technologies | Cat. #A-21245; RRID: AB_2338074 |
| Anti-mouse HRP (goat) | Millipore | Cat. #AP181P RRID: AB_11125547 |
| Anti-rabbit HRP (goat) | Millipore | Cat. #12-348 RRID: AB_11125142 |
| Argonaute-2 (mouse, IP) | Millipore | Cat. #04-642 RRID: AB_10695648 |
| Argonaute-2 (rabbit, WB) | Cell Signaling | Cat. #2897 RRID: AB_2096291 |
| Collybistin (rabbit) | Synaptic Systems | Cat. #261 003; RRID: AB_2619977 |
| GABA$_A$Rα1 (mouse) | NeuroMab | Cat. #75136; RRID: AB_10697873 |
| GABA$_A$Rα5 (mouse) | NeuroMab | Cat. #455 510; RRID: AB_2491075 |
| GABA$_A$Rβ3 (mouse) | NeuroMab | Cat. #75149; RRID: AB_10673389 |
| GABA$_A$Rγ2 (mouse, WB) | NeuroMab | Cat. #75442; RRID: AB_2566822 |
| GABA$_A$Rγ2 (guinea pig, ICC) | Synaptic Systems | Cat. #224 004; RRID: AB_10594245 |
| GAPDH (mouse) | GeneTex | Cat. #627 408; RRID: AB_11174761 |
| GFP (mouse) | NeuroMab | Cat. #75131; RRID: AB_10671444 |
| GluA1 (rabbit) | Millipore | Cat. #ABN 241; RRID: AB_2721164 |
| GluA2 (mouse) | Invitrogen | Cat. #32-0300; RRID: AB_2533058 |
| GPHN (mouse, WB) | Synaptic Systems | Cat. #147 111 RRID: AB_887719 |
| GPHN (mouse, ICC) | Synaptic Systems | Cat. #147 011; RRID: AB_887717 |
| IgG control (mouse) | Cell Signaling | Cat. #5415 RRID: AB_10829607 |
| Neuroligin-2 (mouse) | Synaptic Systems | Cat. #129 511; RRID: AB_2619813 |
| VAMP2 (mouse) | Synaptic Systems | Cat. #104 211; RRID: AB_2619758 |
| VGAT (rabbit) | Synaptic Systems | Cat. #131 003; RRID: AB_887869 |
| VGAT Oyster-650 (rabbit) | Synaptic Systems | Cat. #131 103C5; RRID: AB_2254821 |
| VGlut1 (rabbit) | Synaptic Systems | Cat. #135 302; RRID: AB_887877 |
| **Oligonucleotides and other sequence-based reagents** | | |
| *Cloning Primers* | | |
| *Gphn* 3'UTR SpeI F: TATATAACTAGTTGACTGTATCCTGTCATATGC | This paper | n/a |
| *Gphn* 3'UTR MluI R: CGTATAACGCGTTTTTTAAATAATGATCAAGG | This paper | n/a |
| *Gphn* 3'UTR 153 Mut F: GACTGTATCCTGTCACAGGTATCGGCACAGCTAG | This paper | n/a |
| *Gphn* 3'UTR 153 Mut R: CTAGCTGTGCCGATACCTGTGACAGGATACAGTC | This paper | n/a |
| miR153 Upstream Seq SacI F: TATATA GAGCTC TGCGCAGGACCCAGCAGC | This paper | n/a |
| miR153 Upstream Seq HindIII R: TATATAAAGCTTCTAAGTAGCTGGCAAAGT | This paper | n/a |
| miR153 NFAT Mut F: CACCTCTTGCTGTGTGCGATGCATCCACTAACG | This paper | n/a |
| miR153 NFAT Mut R: CGTTAGTGGATGCATCGCACACAGCAAGAGGTG | This paper | n/a |
| *qPCR Primers* | | |
| miR153 F: TTGCATAGTCACAAAAGTGATC | This paper | n/a |
| miR15a F: TAGCAGCACATAATGGTTT | This paper | n/a |
| miR410 F: AATATAACACAG | This paper | n/a |
| Pri-153 F: AGCGGTGGCCAGTGTCATT | This paper | n/a |
| Pri-153 R: CACAGTTTCCAATGATCAC | This paper | n/a |
| Pri-410 F: TGCTCCGGTCAACACTGGGT | This paper | n/a |
| Pri-410 R: AAAACAGGCCATCTGTGTTA | This paper | n/a |
| *Gphn* F: GGAGACAACCCAGATGACTTAC | This paper | n/a |
| *Gphn* R: CCAGCACCTGCTTGAGATAG | This paper | n/a |
| *Nfatc3* F: TGGCATCAACAGTATGGACCTGGA | This paper | n/a |
| *Nfatc3* R: TTTACCACAAGGAGAAGTGGGCCT | This paper | n/a |
| *Gapdh* F: GATGCTGGTGCTGAGTATGT | This paper | n/a |
| *Gapdh* R: GCTGACAATCTTGAGGGAGTT | This paper | n/a |
| miR153 Upstream Seq F:GGGTTCTAGTCTCGGAACAATAG | This paper | n/a |
| miR153 Upstream Seq R: GGGCTCTGGCAACAGTTAAT | This paper | n/a |
| **Chemicals, Enzymes and other reagents** | | |
| Actinomycin D | Tocris | Cat. #1229 |
| BAPTA-AM | Tocris | Cat. #2787 |
| Cycloheximide | Sigma | Cat. #C7698 |
| Cyclosporin A | Tocris | Cat. #1101 |
| CNQX | Tocris | Cat. #1045 |
| FK506 | Tocris | Cat. #3631 |
| NBQX | Tocris | Cat. #0373 |
| NMDA | Tocris | Cat. #0114 |
| Trichostatin | Tocris | Cat. #1406 |
| TTX | Tocris | Cat. #1078 |
| **Software** | | |
| Prism 10 | GraphPad | https://www.graphpad.com/scientific-software/prism/ |
| ImageJ | NIH | https://imagej.nih.gov/ij/ |
| MiniAnalysis | Synaptosoft | http://www.synaptosoft.com/; RRID:SCR_002184 |
| Axograph X | Axograph | https://axograph.com/download |
| Slidebook 6.0 | 3i | https://www.intelligent-imaging.com/slidebook |

| Reagent/Resource | Reference or Source | Identifier or Catalog Number |
|---|---|---|
| **Other** | | |
| Dual-luciferase reporter assay system | Premega | Cat. #E1910 |
| EpiQuik Acetyl-Histone H3 ChIP kit | EpiGenetek | Cat. #P-2010 |
| miScript II RT kit | Qiagen | Cat. #218160 |
| RNeasy mini kit | Qiagen | Cat. #74104 |
| Immobilon Classico Western HRP substrate | Millipore | Cat. #WBLU0100 |
| Immobilon Crescendo Western HRP substrate | Millipore | Cat. #WBLUR0500 |
| Lipofectamine 2000 | ThermoFisher | Cat. #1166802 |
| Protein G Sepharose | Sigma | Cat. #P3296 |
| miRDIAN miRCon inhibitor | Horizon | Cat. #IN-001005-01-05 |
| miRDIAN miR153 inhibitor | Horizon | Cat. #IH-320381-06-0002 |

## Animals

All animal procedures were conducted in accordance with National Institutes of Health (NIH)–United States Public Health Service guidelines and with the approval of the University of Colorado, Denver, Institutional Animal Care and Use Committee (Protocol # 0426).

## Dissociated hippocampal cultures

Rat hippocampal neurons were dissected from rats (postnatal day 1–2; of mixed sex) and prepared as previously described (Crosby et al, 2019). Briefly, hippocampi were disassociated in papain, and neurons were seeded onto coverslips or dishes coated with poly-D-lysine. Neuron density for 18 mm glass coverslips in 12 well plates was 150,000 and 3,000,000 for 6 cm dishes. Neurons were cultured in Neurobasal media (GIBCO) supplemented with B27 (GIBCO) and 2 mM Glutamax. Cells were maintained at 37 °C, 5% $CO_2$ for 14–18 days before experimental use. Half of the neuronal media was replaced with fresh media and anti-mitotics at DIV5, with a subsequent media change at DIV10.

## Cell lines

HEK293T cells were obtained from ATCC and maintained under standard conditions (10% FBS in DMEM, 37 °C, 5% $CO_2$).

## DNA constructs

Adeno associated viruses (AAVs) overexpressing miR153 or scrambled control (miRCon) were purchased from VectorBuilder and used to infect cultured neurons 4 days prior to in vitro experiments. The miR153 and miRCon overexpression constructs with GFP expression cassette used in imaging experiments were a gift from Danesh Moazed (Mathew et al, 2016). Gephyrin-FingR-GFP was a gift from Dr. Don Arnold (Gross et al, 2013) and Gephyrin-FingR-mScarlet was a gift from Dr. Ulli Bayer (Cook et al, 2019). The 3'UTR of Gphn and the miR153 upstream sequences were created as gene fragments by Twist Bioscience. The *Gphn* 3'UTR was cloned into the SpeI and MluI restriction sites of pMIR-REPORT, and the miR153 seed site was mutated via site directed mutagenesis (*Gphn* 3'UTR SpeI F: TATATAACTAGTTG ACTGTATCCTGTCATATGC, *Gphn* 3'UTR MluI R: CGTAT AACGCGTTTTTTAAATAATGATCAAGG, *Gphn* 3'UTR 153

Mut F: GACTGTATCCTGTCACAGGTATCGGCACAGCTAG, *Gphn* 3'UTR 153 Mut R: CTAGCTGTGCCGATACCTGTGAC AGGATACAGTC). miR153 upstream sequence was cloned into SacI and HindIII sites (miR153 Upstream Seq SacI F: TATATA GAGCTC TGCGCAGGACCCAGCAGC, miR153 Upstream Seq HindIII R: TATATAAAGCTTCTAAGTAGCTGGCAAAGT) of promoter-less pGL4.10 Firefly luciferase construct (Promega), and the NFAT binding site mutated via site directed mutagenesis (miR153 NFAT Mut F: CACCTCTTGCTGTGTGCGATGCATCC ACTAACG, miR153 NFAT Mut R: CGTTAGTGGATGCATCGC ACACAGCAAGAGGTG). NFAT knockdown constructs were generated using oligonucleotides based on previously validated NFATc3 mRNA target sequences (Vashishta et al, 2009; Ulrich et al, 2012), which were then subcloned into pmU6-[shRNA] ('pSilencer'). Cell-specific promoter sequences were not used in any experiments.

## Transfections

DIV 12-14 hippocampal neurons were transfected with plasmid DNA and/or miRDIAN miRNA inhibitors (Horizon) using Lipofectamine 2000 (Invitrogen) and used for experiments at DIV14-16.

## Stereotactic viral injections

Mice bred at the University of Colorado Anschutz from a B6 genetic background were housed in a dedicated animal care facility. This facility was maintained at 35% humidity, 21–23 °C, on a 14/10 light/dark cycle, and mice were housed in ventilated cages with same-sex littermates in groups of 2–5. Food and water were provided ad libitum. All procedures were conducted in accordance with guidelines approved by the Administrative Panel on Laboratory Animal Care at University of Colorado Anschutz, School of Medicine, accredited by the Association for Assessment and Accreditation of Laboratory Animal Care International. Mice were stereotactically injected with miRCon and miR153 OE viruses at P21 and subsequent electrophysiology experiments were performed 3 weeks later to allow time for viral expression. Animals induced at 5% isoflurane were maintained at 1–2% isoflurane and head fixed to the stereotactic frame (Kopf Instruments). Using a drill held in the stereotaxic frame, small holes were made in the skull and 0.5 μL AAV solution was injected into the CA1 hippocampus using a Nanoject III (Drummond Scientific) at a rate of 0.1 μL/min at coordinates (in mm): anteroposterior −3, mediolateral 3.5, and dorsoventral −3.

## Drug treatments

Chemical iLTP was induced in DIV14-18 hippocampal neurons via bath application of 20 μM NMDA and 10 μM CNQX in HBS solution (in mM): 145 NaCl, 2 KCl, 10 HEPES, 2 $CaCl_2$, 2 $MgCl_2$, 10 glucose (pH adjusted to 7.4). iLTP solution was applied for 2 min at 37 °C as previously described (Petrini et al, 2014). In control conditions, neurons were treated for 2 min with HBS (sham) solution at 37 °C. Neurons were imaged live, recorded, harvested, or fixed at multiple time points following stimulation as specified in the figures and/or figure legends. 5 μM cyclosporin A (Tocris), 5 μM FK506 (Tocris), or 20 μM BAPTA-AM (Tocris) were

added to cell media 15 min prior to iLTP induction and left in media post-stimulation throughout live imaging or until harvested. 1 μM trichostatin A (Tocris) was added to cell media 16 h prior to iLTP induction, and 10 μg/mL cycloheximide (Sigma) was added to bath media and conditioned media throughout live imaging.

## RNA isolation and qRT-PCR

RNA was isolated from neurons using the RNeasy mini kit (QIAGEN), and miRNA and mRNA were reverse transcribed into cDNA with the miScript II RT kit (QIAGEN) both according to manufacturer instructions. 1 μL cDNA (diluted 1:10 in RNAase-free water, except when measuring pri-miRNA levels) was used in each qPCR reaction. miRNA-specific primers and the universal primer provided with the miSCript II RT kit or gene-specific primers were used for qPCR in a Biorad CFX384 real-time qPCR system. When quantifying pri- or mature miR153 levels over time, we used miR15a or miR410 expression time course data as controls as we have previously shown that iLTP stimulation does not alter these miRNA levels in neurons (Rajgor et al, 2020). qPCR readings were normalized to the U6 snRNA or *Gapdh* mRNA, with U6 primers included in the miScript II RT kit (QIAGEN) and *Gapdh* primers provided by Integrated DNA Technologies. AGO2-bound mRNAs were normalized to respective inputs. Each qPCR run included 40 cycles with parameters: 94 °C for 15 min, 55 °C for 30 s, and 70 °C for 30 s.

### Primers

miR153 F: TTGCATAGTCACAAAAGTGATC; miR15a F: TAGC AGCACATAATGGTTT; miR410 F: AATATAACACAG; Pri-153 F: AGCGGTGGCCAGTGTCATT; Pri-153 R: CACAGTTTCCAA TGATCAC; Pri-410 F: TGCTCCGGTCAACACTGGGT; Pri-410 R: AAAACAGGCCATCTGTGTTA; *Gphn* F: GGAGACAACCCA GATGACTTAC; *Gphn* R: CCAGCACCTGCTTGAGATAG; *Nfatc3* F: TGGCATCAACAGTATGGACCTGGA; *Nfatc3* R: TTTACCAC AAGGAGAAGTGGGCCT; *Gapdh* F: GATGCTGGTGCTGAGTA TGT; *Gapdh* R: GCTGACAATCTTGAGGGAGTT.

## Luciferase assays

DIV12-14 hippocampal neurons or HEK293T cells plated in 12 well dishes were co-transfected with 500 μg each of Renilla luciferase construct and the appropriate Firefly luciferase reporter construct. miRCon or miR153 were co-transfected when appropriate. Dual-luciferase reporter assays (Promega) were performed according to manufacturer instructions. Briefly, cultured cells were gently rocked in 1x passive lysis buffer for 15 min at room temperature prior to treatment with luciferase assay reagents LARII and Stop & Glo (Promega) to measure Firefly and Renilla luciferase activity, respectively, with the Modulus Microplate Reader (Turner BioSystems) luminometer function. Firefly activity was normalized to Renilla activity.

## Western blotting

Cells were scraped in 2x protein loading buffer: 4% SDS, 20% glycerol, 120 mM Tris at pH 6.8, 0.02% bromophenol blue, 5% 2-mercaptoethanol. Extracts were then boiled for 5 min at 95 °C and proteins were resolved using SDS-PAGE. Wet transfer apparatus was used to transfer proteins to PVDF membranes which were then blocked with 5% milk solution in PBS-Tween. Membranes were

incubated at 4 °C overnight with the appropriate primary antibody: GPHN (1:5000 Synaptic Systems 147111), GABA$_A$Rα1 (1:1000 NeuroMab 75136), GABA$_A$Rγ2 (1:1000 NeuroMab 75442), GAPDH (1:10,000 GeneTex 627408), GFP (1:2000 NeuroMab 75131), GABA$_A$R β3 (1:1000 NeuroMab 75149), GABA$_A$Rα5 (1:1000 NeuroMab 455510), GluA1 (1:1000 Millipore ABN241), NL2 (1:2500 Synaptic Systems 129511), CB (1:1000 Synaptic Systems 261 003), VAMP2 (1:5000 Synaptic Systems 104211). Membranes were then washed with PBS-Tween and incubated with the appropriate HRP conjugated secondary antibody for 1 h at room temperature (1:10,000 Millipore). When appropriate, membranes were stripped with stripping buffer and re-probed. ECL western blotting substrates (Millipore) were used to visualize protein bands and images were analyzed using ImageJ to obtain densitometry measurements. The integrated density of the protein band of interest was normalized to that of GAPDH in the same gel.

## Immunofluorescence microscopy

For surface staining, hippocampal neurons on coverslips were fixed with a 4% paraformaldehyde/4% sucrose solution in PBS for 5 min at room temperature, followed by PBS washes. After incubating in blocking solution (3% bovine serum albumin, 2% normal goat serum in PBS) for 40 min at room temperature, coverslips were surface stained for GABA$_A$Rγ2 (1:500, Synaptic Systems 224004) or GluA2 (1:200, Invitrogen 32-0300). After surface labeling, neurons were permeabilized with 0.5% NP-40 for 2–3 min and blocked for 40 min prior to GPHN (1:500, Synaptic Systems 147011), VGAT (1:1000, Synaptic Systems 131003), or VGluT1 (1:1000, Synaptic Systems 135302) staining. All primary antibodies were diluted in blocking solution, and incubations were as follows: GABA$_A$Rγ2, GPHN, and VGAT for 1 h at room temperature; GluA2 for 3 h at room temperature; VGluT1 overnight at 4 °C. Coverslips were then washed 3× 5 min in PBS and incubated with appropriate secondary antibodies for 1 h at room temperature (1:1000, ThermoFisher Alexa-Fluor 488, 568, 647). After mounting coverslips onto microscope slides with ProLong gold mounting media (Thermo-Fisher), confocal images were acquired using: an Axio Observer microscope (Zeiss) equipped with a Tokogawa CSU-X1 spinning disk unit, an EC Plan-Neofluar 63x Plan-Apo (1.4 NA) oil immersion objective lens, a Photometrics Evolve 512 EMCCD camera with 16-bit dynamic range, and SlideBook 6.0 software (3i). Images were acquired using 488, 561, and 640 nm laser excitation to capture 3.96 μm z-stacks (13 xy planes, 0.33 μm intervals) in each channel. The maximum intensity of these planes was projected onto 2D images. The cluster area and density of fluorescent puncta in randomly selected dendrites were analyzed using ImageJ, based on a minimum cluster area threshold of 0.05 μm².

## Whole-cell electrophysiology

### In vitro

DIV16-20 pyramidal hippocampal neurons expressing GFP were patched in whole-cell mode in an extracellular ACSF solution containing (in mM): 10 HEPES, 130 NaCl, 5 KCl, 30 D-glucose, 2 CaCl₂, and 1 MgCl₂ equilibrated with 95% O₂/5% CO₂. mIPSCs were isolated using 2 μM TTX, 100 μM APV, and 10 μM NBQX. Dissociated neurons were held at −70 mV (no treatments) or −65 mV (sham/iLTP treatments) 2 min prior to recording 2 min

of continuous activity using an internal solution containing (in mM): 67.5 CsCl, 67.5 CsMeSO$_4$, 0.1 CaCl$_2$, 2 MgCl$_2$, 10 HEPES, 0.1 EGTA, 0.5 Na$_3$GTP, 3 Na$_2$ATP, 10 phosphocreatine (pH adjusted to 7.25 with CsOH). In iLTP experiments, treatments were administered as described above. mIPSCs were recorded under basal conditions from multiple different GFP-expressing neurons and the average amplitude and frequency were considered control values. For the iLTP condition, neurons from the same experimental culture were stimulated and mIPSCs recorded approximately 45–60 min later. mIPSC parameters were steady state at this phase of iLTP and the average amplitude and frequencies across the recording were used as iLTP values.

Data were collected with a Multiclamp 700b amplifier and digitized with a National Instruments DAQ board at 10 kHz (filtered at 2 kHz, single pole Bessel filter). mIPSCs were quantified using MiniAnalysis software (Synaptosoft Inc.).

### Ex vivo

Animals anesthetized with isoflurane were decapitated and their brains rapidly dissected. Horizontal slices (240 μm) were sectioned with a vibratome (Leica VT1200) in cutting solution (in mM): 75 NaCl, 2.5 KCl, 6 MgCl$_2$, 0.1 CaCl$_2$, 1.2 NaH$_2$PO$_4$, 25 NaHCO$_3$, 2.5 D-glucose, 50 sucrose. Slices were then incubated for 30 min in 31.5 °C oxygenated ACSF (in mM): 126 NaCl, 2.5 KCl, 1.2 MgCl$_2$, 2.5 CaCl$_2$, 1.2 NaH$_2$PO$_4$, 21.4 NaHCO$_3$, 11.1 D-glucose, also with 10 μM DNQX and 0.5 μM TTX. Tissue recovered at room temperature for at least 1 h prior to whole-cell patch clamp recordings. Cells were voltage-clamped at −70 mV using 4–6 MΩ patch pipettes in 29.5 °C ACSF. All recordings were acquired with Axopatch 200B Amplifiers (Molecular Devices) and Axograph X (Axograph Scientific). mIPSC analysis was performed with Axograph X.

### Live imaging

Cultured hippocampal neurons were transfected with Gephyrin-FingR-GFP or co-transfected with Gephyrin-FingR-mScarlet and GFP-tagged miRNA constructs 48–72 h prior to live imaging experiments. Coverslips were incubated with VGAT-Oyster$^{650}$ antibody (1:200, Synaptic Systems 131103C5) in conditioned media for 20 min at 37 °C. After mounting coverslips in a Ludin chamber with 37 °C HBS solution, transfected pyramidal neurons were imaged pre-stimulation ($t = 0$ min) and then at multiple timepoints 10–90 min following drug treatments in warmed HBS. Confocal images were acquired using the same spinning disk confocal equipment described for fixed immunofluorescent imaging. Images were acquired using 488, 561, and 640 nm laser excitation to capture 3.96 μm z-stacks (13 xy planes, 0.33 μm intervals) in each channel. The maximum intensity of these planes was projected onto 2D images. Fluorescent GPHN puncta co-localized with VGAT were randomly selected in dendrites and their background-subtracted mean fluorescence intensity at each timepoint was measured with ImageJ. The cluster area of VGAT and density of GPHN and VGAT fluorescent puncta in randomly selected dendrites were analyzed using ImageJ, based on a minimum cluster area threshold of 0.05 μm$^2$.

### Chromatin immunoprecipitation (ChIP)

Acetyl-Histone H3 ChIP experiments utilized the EpiQuik Acetyl-Histone H3 ChIP kit (EpiGenetek) and were performed according to manufacturer instructions. Briefly, cells were in vivo crosslinked using 1% formaldehyde, then lysed and DNA sheared via lysis buffer treatment and sonication. After aliquoting input DNA, the

remaining DNA was incubated with Non-immune IgG (negative control) or Anti-Acetyl-Histone H3 antibody for 1 h at room temperature. Next, all samples were treated with Proteinase K in DNA Release Buffer (65 °C, 15 min) and DNA Cross-linking Reverse Buffer (65 °C, 90 min), purified via wash and spin steps, and finally eluted. DNA eluted from the IP was normalized to input DNA with qPCR using forward and reverse primers (miR153 Upstream Seq F: GGGTTCTAGTCTCGGAACAATAG, miR153 Upstream Seq R: GGGCTCTGGCAACAGTTAAT) targeting the miR153 upstream sequence.

### AGO2 RNA immunoprecipitation (IP)

Hippocampal neurons were lysed in 1 mL lysis buffer (in mM, unless otherwise specified): 10 HEPES at pH 7.4, 200 NaCl, 30 EDTA, 0.5% Triton X-100, 0.5 U/μL SUPERase inhibitor, 1x Complete Protease Inhibitor. Lysates were then briefly sonicated for 10 s and left on ice for 10 min. Cell debris was pelleted with centrifugation (13,000 rpm for 10 min, 4 °C) and 50 μL was taken as protein input while the remaining lysate was pre-cleared using 100 μL or Protein G Sepharose beads (Sigma) at 4 °C for 30 min. After pelleting the beads, remaining lysate was equally divided and incubated with 2 μg Pan-Ago (Millipore; 2A8) or IgG (Cell Signaling 5415) at 4 °C for 1 h. Lysates were incubated with beads at 4 °C for 1 h to capture antibody complexes, and then washed 5 times in 1 mL of lysis buffer. 25% of the purified complex was boiled in protein loading buffer followed by western blotting to quantify AGO2 enrichment, and the remaining immunoprecipitate was used for RNA isolation.

### Quantification and statistical analysis

Every experiment was performed at least three times, with experimenter blind to experimental conditions (when possible), to ensure rigor and reproducibility. Sample size was based on prior studies. Independent experiments were performed on separate neuronal preparations ($N = 3$), with the exception of the iLTP electrophysiology experiments (Fig. 6) recorded on 3 separate days, but two of which were measured on consecutive days from the same culture ($N = 2$). All statistical tests were performed in Prism10 (GraphPad). Raw value data (Figs. 2, 3, 5, EV2, EV3) were assessed for normality using the D'Agostino-Pearson test or the Shapiro–Wilk test for small sample sizes ($n < 7$). When comparing two groups with a single variable, t-tests (normal data) or Mann–Whitney tests (non-normal data) were used to determine statistical significance. When comparing 3 or more groups with a single variable, one-way ANOVA (parametric) or Kruskal–Wallis (non-parametric) tests were used. For unpaired fold change measurements without control error-correction (Figs. 1, 4, EV1, EV4), one-sample t-tests (acceptable data skewness/kurtosis, no outliers) or Wilcoxon signed rank tests were used to compare experimental values against the hypothetical value (1.0) to determine statistical significance. In experiments testing 2 variables (Figs. 1, 4–7), statistical significance was determined using ordinary two-way ANOVA with post-hoc multiple comparison tests.

For experiments in which multiple neurons ("n") from a single culture ("N") were analyzed (Figs. 2, 3, 5, 7, EV2, EV3, EV5), data was organized into clusters based on their N. In such experiments comparing two groups (Figs. 2, 3, EV2, EV3), normal clustered data were analyzed with the nested t-test. Quantification graphs of unpaired clustered data include individual data points (neurons) represented as different symbols based on neuronal preparation. Live imaging experiments entailed repeated measures of the same neurons over

time (Figs. 5, 7, EV5). Statistical significance of these paired data was determined with mixed effects analysis and post-hoc multiple comparison tests. Exact sample sizes and statistical tests used to determine significance for each experiment are specified in the figure legends. *P*-values were considered significant if <0.05. Error bars on all quantification graphs represent the standard error of the mean (SEM).

## Data availability

The raw data of this publication have been deposited in the BioImage Archive database (https://www.ebi.ac.uk/bioimage-archive/) and assigned the unique identifier: https://doi.org/10.6019/S-BIAD1266.

The source data of this paper are collected in the following database record: biostudies:S-SCDT-10_1038-S44319-024-00253-z.

## Peer review information

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

## Acknowledgements

This work was supported by NIMH grants R01MH119154 and R01MH128199 to KRS, NINDS institutional research training grant T32NS099042 and Ruth L. Kirschstein individual predoctoral National Research Service Award (NRSA) F31NS12410 to TMW, NINDS Blueprint Diversity Specialized Predoctoral to Postdoctoral Advancement in Neuroscience (D-SPAN) Award (FNS120640A) to JDG, NINDS individual predoctoral NRSA F31NS130979 to DJK, NIDA individual predoctoral NRSA for MD/PhD (F30DA048543) to SMZ, NIMH grant R01MH123700 to MLD, NINDS grant R35NS116879 to MJK, NIDA grant R01DA35821 and NINDS grant R01NS95809 to CPF. Contents are the authors' sole responsibility and do not necessarily represent official NIH views.

## Author contributions

**Theresa M Welle**: Data curation; Formal analysis; Funding acquisition; Writing —original draft; Writing—review and editing. **Dipen Rajor**: Conceptualization; Data curation; Formal analysis; Investigation. **Dean J Kareemo**: Formal analysis; Investigation. **Joshua D Garcia**: Data curation; Formal analysis; Investigation. **Sarah M Zych**: Data curation; Formal analysis; Investigation. **Sarah E Wolfe**: Formal analysis; Investigation. **Sara E Gookin**: Formal analysis; Investigation. **Tyler P Martinez**: Investigation. **Mark L Dell'Acqua**: Supervision; Funding acquisition. **Christopher P Ford**: Supervision; Funding acquisition; Investigation. **Matthew J Kennedy**: Supervision; Funding acquisition. **Katharine R Smith**: Conceptualization; Data curation; Formal analysis; Supervision; Funding acquisition; Investigation; Project administration; Writing—review and editing.

Source data underlying figure panels in this paper may have individual authorship assigned. Where available, figure panel/source data authorship is listed in the following database record: biostudies:S-SCDT-10_1038-S44319-024-00253-z.

## Disclosure and competing interests statement

The authors declare no competing interests.

# Expanded View Figures

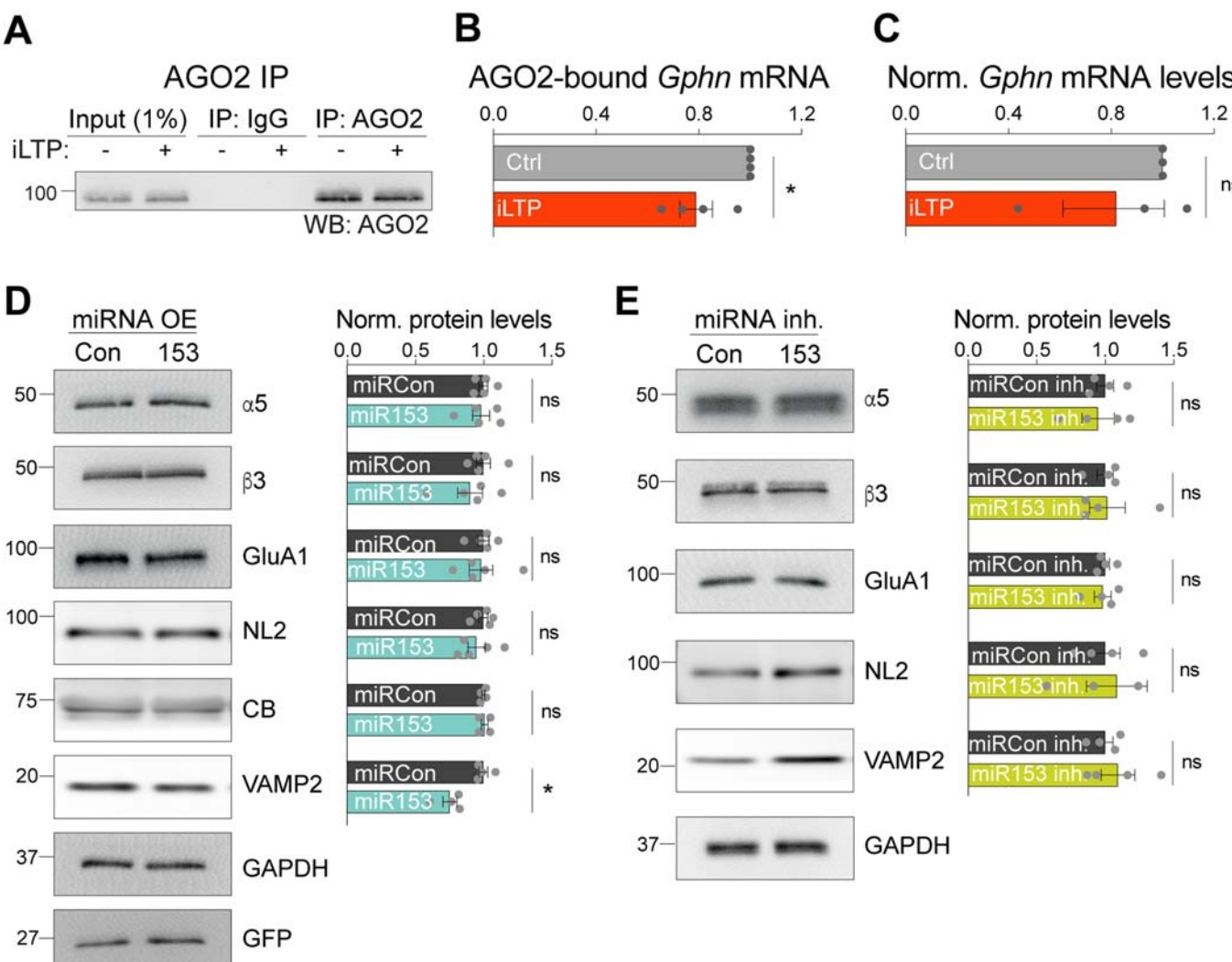

**Figure EV1. Control experiments for AGO2 IPs and impact of miR153 on other synaptic proteins.**

(A) Western blot (WB) of AGO2 immunoprecipitated from neurons following control treatment (Ctrl) or 90 min post-iLTP stimulation (iLTP). (B) qRT-PCR of *Gphn* mRNA bound to AGO2 in neurons from (A). AGO2-bound *Gphn* was normalized to total *Gphn* mRNA expression, and fold change from Ctrl was quantified for each condition. $N = 4$. $P = 0.0452$. (C) qRT-PCR of total *Gphn* mRNA levels in Ctrl and iLTP-90 neurons. *Gphn* mRNA levels were normalized to U6 expression, and fold change from Ctrl was quantified for each condition. $N = 3$. $P = 0.7500$. (D) Left: western blots of GABA$_A$R subunits α5 (extrasynaptic) and β3 (synaptic), AMPAR subunit GluA1, GPHN binding proteins neuroligin-2 (NL2) and collybistin (CB), miR153 target VAMP2, GAPDH, and GFP protein levels in neurons overexpressing miRCon or miR153miRNA overexpression (OE) constructs contain a GFP reporter. Right: quantification of α5, β3, GluA1, NL2, CB, VAMP2 in miRCon or miR153 OE neurons. Protein levels were normalized to GAPDH, and the data quantified as relative change in normalized protein expression. $N = 5$. P-values (miRCon vs miR153): α5 > 0.9999, β3 = 0.5476, GluA1 = 0.5476, NL2 = 0.4206, CB > 0.9999, VAMP2 = 0.0286. (E) Left: western blots of α5, β3, GluA1, NL2, VAMP2, and GAPDH protein levels in neurons expressing miRCon or miR153 inhibitors. Right: quantification of α5, β3, GluA1, NL2, CB, VAMP2 in miRCon neurons or neurons in which miR153 was inhibited. Protein levels were normalized to GAPDH, and the data quantified as relative change in normalized protein expression. $N = 4$. P-values (anti-Con vs anti-153): α5 = 0.8857, β3 = 0.8857, GluA1 = 0.8857, NL2 = 0.8857, CB > 0.9999, VAMP2 = 0.6857. $N =$ independent neuronal cultures/experiments. All values represent mean ± SEM. *$p < 0.05$, **$p < 0.01$, ***$p < 0.005$, ****$p < 0.0001$; one-sample t-test (B), Wilcoxon signed rank test (C), and Mann–Whitney test (D, E).

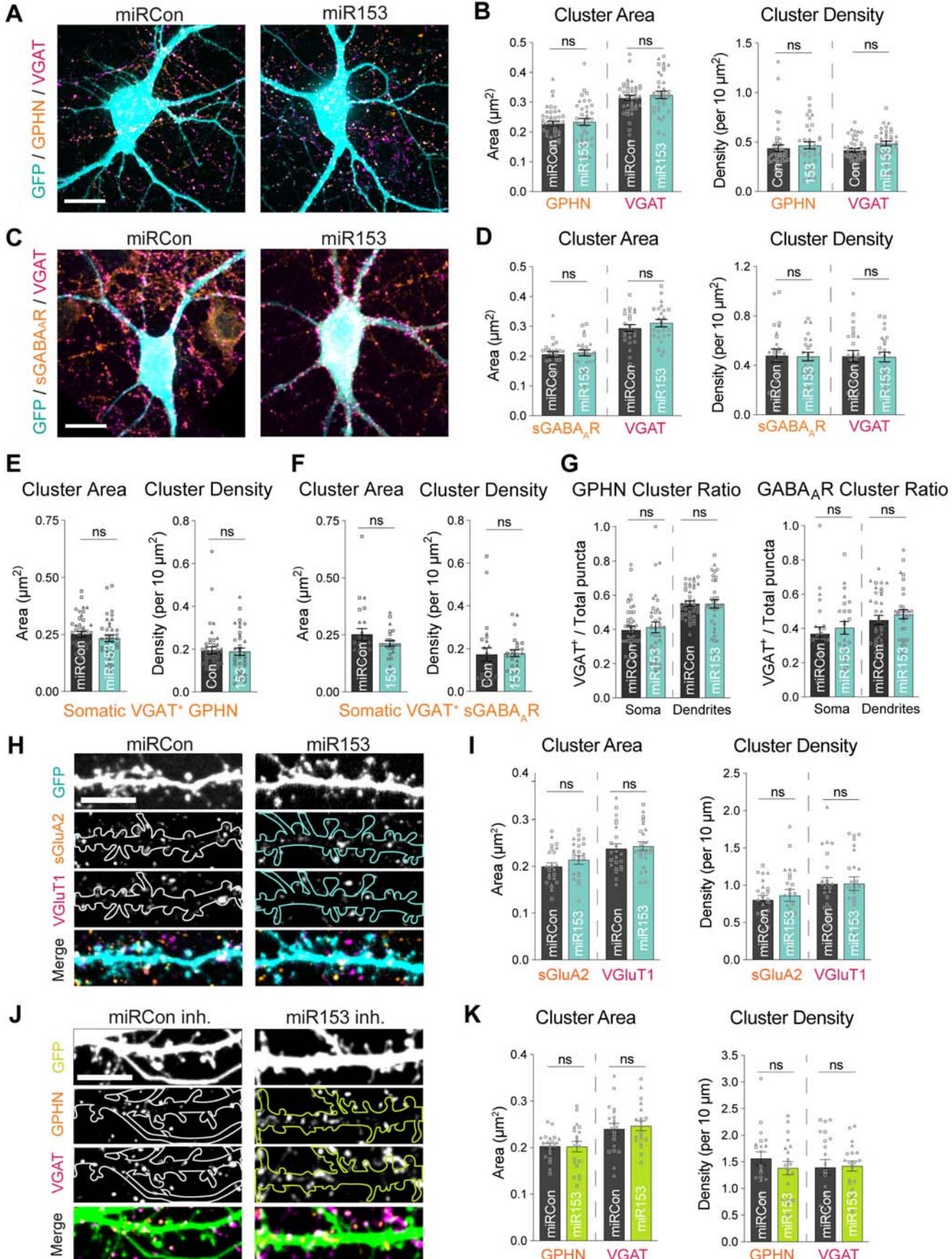

◄ **Figure EV2.  Control experiments for impact of miR153 manipulations on inhibitory somatic synapses and glutamatergic synapses.**

(A) Representative somata of miRCon or miR153 OE-expressing neurons labeled with antibodies to GPHN and VGAT. Scale bar, 80 μm. (B) Quantification of GPHN and VGAT cluster area (left) and cluster density (right) in neurons from (A). $N = 3 / n = 37$–43 neurons per condition. *P*-values (miRCon vs miR153): GPHN area = 0.7738, VGAT area = 0.6588, GPHN density = 0.7784, VGAT density = 0.3919. (C) Representative somata of miRCon or miR153 OE-expressing neurons labeled with antibodies to surface $GABA_AR$ γ2 subunit ($sGABA_AR$) and VGAT. Scale bar, 80 μm. (D) Quantification of $sGABA_AR$ and VGAT cluster area (left) and cluster density (right) in neurons as shown in (C). $N = 3 / n = 32$–35 neurons in each condition. *P*-values (miRCon vs miR153): γ2 area = 0.7593, VGAT area = 0.4208, γ2 density = 0.9159, VGAT density = 0.9221. (E) Quantification of VGAT[+] GPHN cluster area (left) and density (right) from total GPHN puncta quantified in (B). $N = 3 / n = 37$–43 neurons per condition. *P*-values (miRCon vs miR153): GPHN area = 0.3569, GPHN density = 0. 8065. (F) Quantification of VGAT[+] $sGABA_AR$ cluster area (left) and density (right) from total $sGABA_AR$ puncta quantified in (D). $N = 3 / n = 32$-35 neurons per condition. *P*-values (miRCon vs miR153): γ2 area = 0.3028, γ2 density = 0.8555. (G) Proportion of VGAT[+] GPHN clusters (left) and VGAT[+] $sGABA_AR$ clusters (right) in soma and dendrites. $N = 3 / n = 32$–42 neurons. *P*-values (miRCon vs miR153): somatic GPHN = 0.7279, dendritic GPHN = 0.9519, somatic γ2 = 0.5695, dendritic γ2 = 0.3768 (H) Representative dendritic segments of miRCon or miR153 OE neurons labeled with antibodies to surface AMPAR subunit GluA2 (sGluA2) and VGluT1. Scale bar, 10 μm. (I) Quantification of surface GluA2 and VGluT1 cluster area (left) and cluster density (right) in neurons from (H). $N = 3 / n = 23$–24 neurons per condition. *P*-values (miRCon vs miR153): sGluA2 area = 0.2599, VGluT1 area = 0.7703, sGluA2 density = 0.5529, vGluT1 density = 0.9867. (J) Representative dendritic segments of miRCon or miR153 inhibitor-expressing neurons labeled with antibodies to gephyrin (GPHN) and VGAT. Scale bar, 10 μm. (K) Quantification of GPHN and VGAT cluster area (left) and cluster density (right) in neurons from (A). $N = 4 / n = 20$ neurons per condition. *P*-values (anti-Con vs anti-153): GPHN area = 0.9630, VGAT area = 0.7950, GPHN density = 0.5259, VGAT density = 0.9461. $N$ = independent neuronal cultures/experiments, $n$ = neurons. All values represent mean ± SEM. Neurons from different culture preparations are represented by different symbols of data points. \**p* < 0.05, \*\**p* < 0.01, \*\*\**p* < 0.005, \*\*\*\**p* < 0.0001; nested *t*-test.

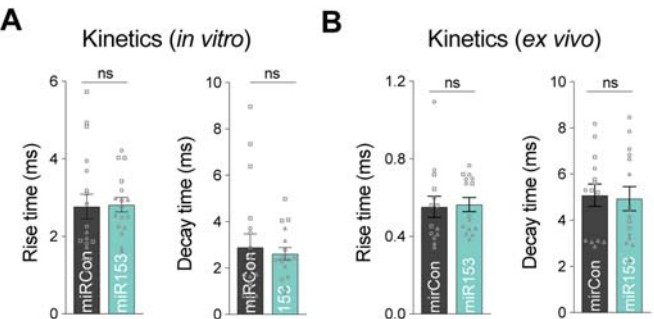

**Figure EV3.  Measurement of mIPSC kinetics.**

(**A**) Quantification of mIPSC rise time (left) and decay time (right) from miRCon and miR153 OE-expressing neurons in culture. $N = 3$ / $n = 17$–18 neurons per condition. *P*-values (miRCon vs miR153): rise time = 0.7836, decay time = 0.9860 (**B**) Quantification of mIPSC rise time (left) and decay time (right) from miRCon and miR153 OE-expressing neurons in slice. $N = 3$–4 / $n = 17$–22 neurons per condition. *P*-values (miRCon vs miR153): rise time = 0.6531, decay time = 0.8593. $N$ = independent neuronal cultures/experiments, $n$ = neurons. All values represent mean ± SEM. Neurons from different neuronal preparations are represented by different symbols of data points. *$p < 0.05$, **$p < 0.01$, ***$p < 0.005$, ****$p < 0.0001$; nested t-test (**A**) and Mann–Whitney test (**B**).

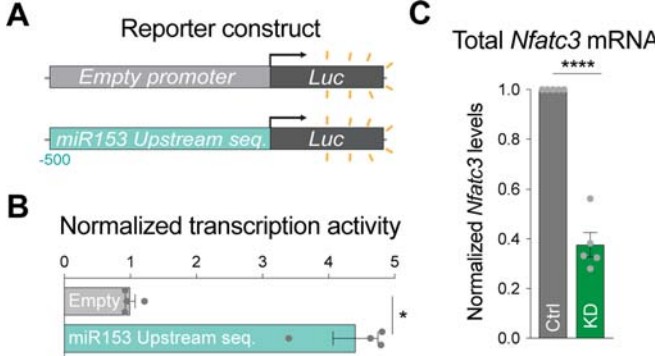

**Figure EV4. Control experiments for Luc reporter construct and NFATc3 knockdown construct.**

(A) Schematic of the Empty-Luc (no promoter) and miR153$^{-500}$-Luc luciferase reporters, designed to test transcriptional activity of the sequence 500 bp upstream of miR153. (B) Quantification of Luc activities in neurons expressing reporters containing no promoter (Empty) or the sequence upstream of pri-miR153 coding region (miR153 Upstream seq.). Firefly was normalized to Renilla, and the data quantified as relative change in normalized Luc activity with error-corrected control values. $N = 4$. $P = 0.0286$. (C) qRT-PCR of total *Nfatc3* mRNA levels in Ctrl and NFATc3 knockdown (NFAT KD) neurons. *Nfatc3* mRNA levels normalized to *Gapdh* mRNA expression, and fold change from Ctrl was quantified for each condition. $N = 6$. $P = 0.0002$. $N =$ independent neuronal cultures/experiments. All values represent mean ± SEM. *$p < 0.05$, **$p < 0.01$, ***$p < 0.005$, ****$p < 0.0001$; Mann–Whitney test (B) and one-sample t-test (C).

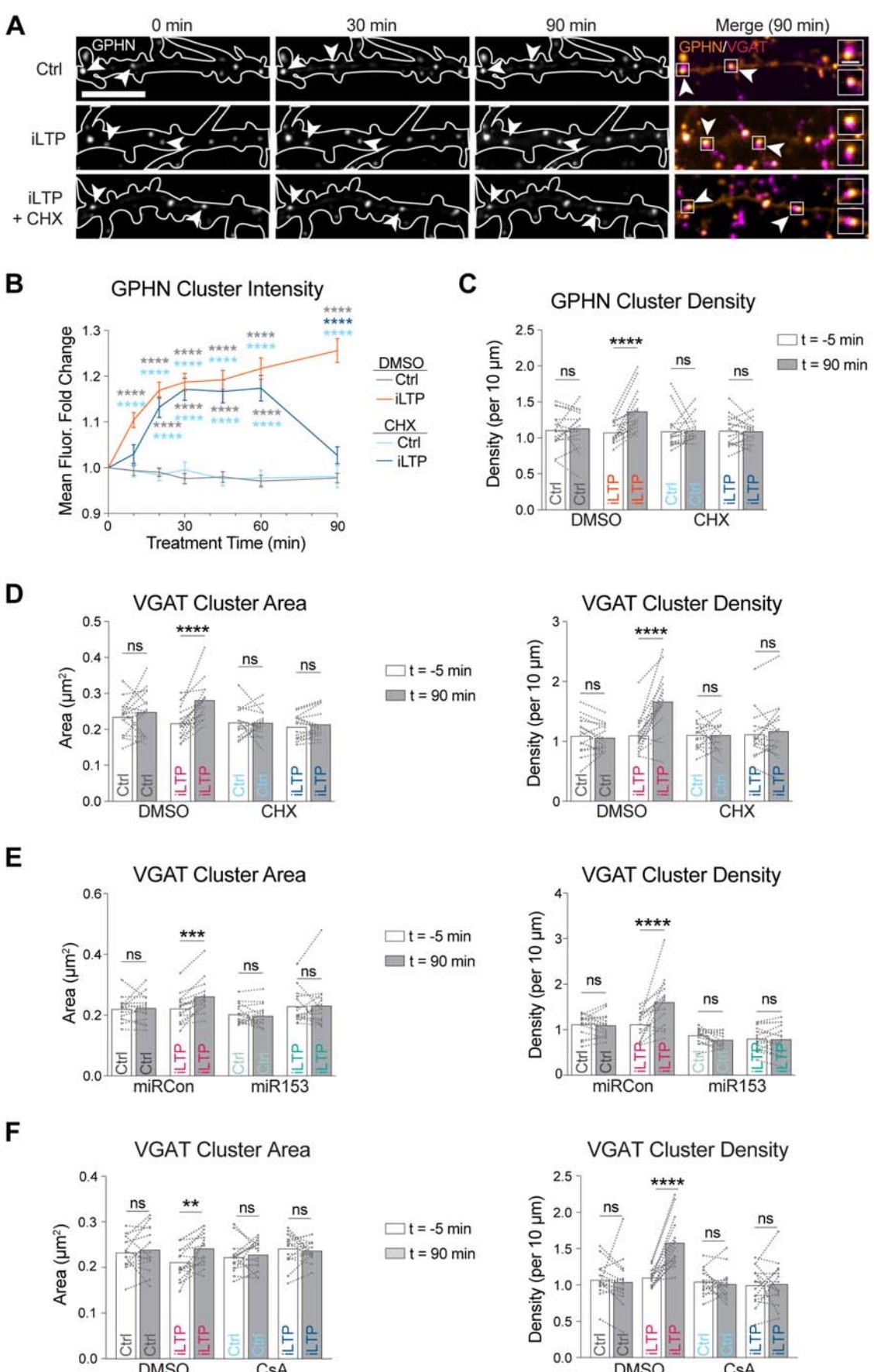

**Figure EV5.   Control experiments for inhibitory synapse live imaging.**

(A) Representative dendritic segments of neurons expressing GPHN IB and labeled with an antibody to VGAT, imaged over time in control and iLTP conditions in the presence or absence of translational inhibitor cycloheximide (CHX). Puncta are labeled with filled arrowheads when the fluorescence is unchanged and open arrowheads when fluorescence increases over time. Boxes indicate the fluorescent puncta enlarged in the merged images (dendrite scale bar, 10 μm; synapse scale bar, 2 μm). (B) Quantification of fold change in GPHN puncta fluorescence intensity over time following treatment in neurons from (A)). $N = 3 / n = 15$ neurons per condition. P-values: 10 min DMSO iLTP vs DMSO Ctrl/CHX Ctrl <0.0001; 20 min DMSO iLTP vs DMSO Ctrl/CHX Ctrl <0.0001, CHX iLTP vs DMSO Ctrl/CHX Ctrl <0.0001; 30 min DMSO iLTP vs DMSO Ctrl/CHX Ctrl <0.0001, CHX iLTP vs DMSO Ctrl/CHX Ctrl <0.0001; 45 min DMSO iLTP vs DMSO Ctrl/CHX Ctrl <0.0001, CHX iLTP vs DMSO Ctrl/CHX Ctrl <0.0001; 60 min DMSO iLTP vs DMSO Ctrl/CHX Ctrl <0.0001, CHX iLTP vs DMSO Ctrl/CHX Ctrl <0.0001; 90 min DMSO iLTP vs DMSO Ctrl/CHX Ctrl/CHX iLTP <0.0001. (C) Paired measurements of GPHN cluster density in dendrites prior to ($-5$ min) and 90 min following treatment. $N = 3 / n = 15$ neurons per condition. P-values ($t = -5$ min vs $t = 90$ min): GPHN density DMSO Ctrl = 0.9874, GPHN density DMSO iLTP <0.0001, GPHN density CHX Ctrl = 0.9996, GPHN density CHX iLTP >0.9999. (D) Paired measurements of VGAT cluster area (left) and density (right) in dendrites prior to ($-5$ min) and 90 min following treatment. $N = 3/n = 15$ neurons per condition. P-values ($t = -5$ min vs $t = 90$ min): VGAT area DMSO Ctrl = 0.6649, VGAT area DMSO iLTP <0.0001, VGAT area CHX Ctrl = 0.9999, VGAT area CHX iLTP = 0.9557; VGAT density DMSO Ctrl = 0.9893, VGAT density DMSO iLTP <0.0001, VGAT density CHX Ctrl >0.9999, VGAT density CHX iLTP = 0.9256. (E) Paired measurements of VGAT cluster area (left) and density (right) in miRCon or miR153 OE neurons (as seen in Fig. 5A) prior to ($-5$ min) and 90 min following treatment. $N = 3 / n = 15$ neurons per condition. P-values ($t = -5$ min vs $t = 90$ min): VGAT area miRCon Ctrl = 0.9966, VGAT area miRCon iLTP = 0.0004, VGAT area miR153 Ctrl = 0.9684, VGAT area miR153 iLTP = 0.9991; VGAT density miRCon Ctrl = 0.9975, VGAT density miRCon iLTP < 0.0001, VGAT density miR153 Ctrl = 0.5521, VGAT density miR153 iLTP > 0.9999. (F) Paired measurements of VGAT cluster area (left) and density (right) in treated neurons (as seen in Fig. 6D) prior to ($-5$ min) and 90 min following treatment. $N = 3/n = 15$ neurons per condition. P-values ($t = -5$ min vs $t = 90$ min): VGAT area DMSO Ctrl = 0.9176, VGAT area DMSO iLTP = 0.0032, VGAT area CsA Ctrl = 0.9442, VGAT area CsA iLTP = 0.9715; VGAT density DMSO Ctrl = 0.9176, VGAT density DMSO iLTP < 0.0001, VGAT density CsA Ctrl = 0.9840, VGAT density CsA iLTP = 0.9986. $N =$ independent neuronal cultures/experiments. All values represent mean ± SEM. *$p < 0.05$, **$p < 0.01$, ***$p < 0.005$, ****$p < 0.0001$; mixed-effects model with Geisser-Greenhouse correction (B) and Šidák's multiple comparisons post-hoc test (B–F).

