## [Peer Review File · EMBO Reports]

miRNA-mediated control of gephyrin synthesis drives sustained inhibitory synaptic plasticity

Theresa Welle, Dipen Rajgor, Dean Kareemo, Joshua Garcia, Sarah Zych, Sarah Wolfe, Sara Gookin, Tyler Martinez, Mark Dell'Acqua, Christopher Ford, Matthew Kennedy, and Katharine Smith

Corresponding author(s): Katharine Smith (katharine.r.smith@cuanschutz.edu)

Review Timeline:

Submission Date:	15th Dec 23
Editorial Decision:	18th Jan 24
Revision Received:	28th Jun 24
Editorial Decision:	16th Aug 24
Revision Received:	22nd Aug 24
Accepted:	2nd Sep 24

Editor: Esther Schnapp / Martina Rembold

Transaction Report:

Dear Dr. Smith,

Thank you for the submission of your manuscript to EMBO reports. We have now received the full set of referee reports as well as referee cross-comments that are all pasted below.

As you will see, the referees acknowledge that the findings are interesting, novel and of good quality. However, they also have several suggestions for how the study can be improved, and I think all should be addressed, except point 3 by referee 3, these *in vivo* experiments do not need to be performed for the publication of your study here. Please let me know in case you have any questions or comments regarding the revisions, and we can discuss this further, also in a video chat, if you like.

I would thus like to invite you to revise your manuscript with the understanding that the referee concerns must be fully addressed and their suggestions taken on board. Please address all referee concerns in a complete point-by-point response. Acceptance of the manuscript will depend on a positive outcome of a second round of review. It is EMBO reports policy to allow a single round of major revision only and acceptance or rejection of the manuscript will therefore depend on the completeness of your responses included in the next, final version of the manuscript.

We realize that it is difficult to revise to a specific deadline. In the interest of protecting the conceptual advance provided by the work, we recommend a revision within 3 months (19th Apr 2024). Please discuss the revision progress ahead of this time with the editor if you require more time to complete the revisions.

- 1) A data availability section providing access to data deposited in public databases is missing. If you have not deposited any data, please add a sentence to the data availability section that explains that.
- 2) Your manuscript contains statistics and error bars based on $n=2$. Please use scatter blots in these cases. No statistics should be calculated if $n=2$.

3) We replaced Supplementary Information with Expanded View (EV) Figures and Tables that are collapsible/expandable online. A maximum of 5 EV Figures can be typeset. EV Figures should be cited as 'Figure EV1, Figure EV2' etc... in the text and their respective legends should be included in the main text after the legends of regular figures.

5) a complete author checklist, which you can download from our author guidelines <https://www.embopress.org/page/journal/14693178/authorguide>. Please insert information in the checklist that is also reflected in the manuscript. The completed author checklist will also be part of the RPF.

6) Please note that all corresponding authors are required to supply an ORCID ID for their name upon submission of a revised manuscript (<https://orcid.org/>). Please find instructions on how to link your ORCID ID to your account in our manuscript tracking system in our Author guidelines <https://www.embopress.org/page/journal/14693178/authorguide#authorshipguidelines>

I look forward to seeing a revised form of your manuscript when it is ready.

Esther Schnapp, PhD

Referee #1:

In this manuscript by Welle et al., a novel miRNA-regulated pathway is described that mediates long-term potentiation (iLTP) of inhibitory synapses. In previous work (Rajgor et al., Cell Reports 2020), this group showed that iLTP is dependent on translation of synaptic proteins and showed that miRNA376c controls translation of specific GABA receptor subunits. Here, using diverse novel molecular tools, biochemistry and microscopy approaches, Welle et al., identify miR153 as a regulator of the translation of Gephyrin, the scaffold protein at inhibitory synapses that is critical for GABA receptor clustering. Specifically, they demonstrate that iLTP induction leads to the gradual downregulation of miR153 expression. Using luciferase reporter assays in several cellular systems, it is then shown that miR153 is able to downregulate Gephyrin mRNA translation by binding to the 3'UTR of Gephyrin mRNA. Consistently, overexpression of miR153 expression reduced protein levels of Gephyrin, but not of other synaptic proteins, and decreased synaptic clustering of Gephyrin and inhibitory synaptic transmission. Furthermore, it is shown that iLTP induction leads to the rapid transcriptional repression of miR153 expression that is controlled by calcineurin, NFATc3 and HDACs. Live-imaging experiments show that the gradual increase in Gephyrin and VGAT synaptic clusters after iLTP induction is abrogated in miR153-overexpression neurons, as well as in neurons treated with CaN blockers. These experiments thus indicate that iLTP leads to the reduced expression of miR153 through CaN/NFATc3/HDAC signaling, effectively relieving the translational repression of Gephyrin mRNA and promoting the translation of Gephyrin and clustering at synaptic sites - ultimately to potentiate inhibitory synaptic transmission.

Altogether, this is a very interesting, complete and rigorous study demonstrating a novel regulatory pathway underlying the regulation of iLTP. The experiments are well-controlled, properly analyzed and clearly presented. The manuscript is clearly written and has the appropriate length and format. This work is likely of broad interest for biologists, bringing important new mechanistic insight in a process that is relevant for brain functions such as learning and memory.

Comments

- All the experiments are consistent with the model that miR153 downregulation is required for iLTP. This is mostly based on confocal microscopy data that indeed miR153 overexpression prevents the iLTP-induced increase in the size and number of inhibitory synapses. Physiological evidence that preventing miR153 downregulation prevents potentiation of inhibitory synapses would strengthen the manuscript. For instance, a simple prediction would be that overexpression of miR153 prevents the potentiation of mIPSC frequency.
- There is a striking difference in mIPSC frequency between the dissociated cultures and ex vivo slices. This could be very well explained by a difference in the number of inhibitory neurons in the different preparations. Nevertheless, the kinetics of mIPSCs quantified in Sup Figure 3 are also very different. How do the authors explain this?
- It would be helpful if the description in the methods section for the luciferase and ChIP procedures were a bit more elaborate. How many cells are used, buffers, instruments etc?
- In the methods section: 'Acetyl-histone H3 chromatic immunoprecipitation' should be 'chromatin immunoprecipitation'.
- In some figure legends 'n' is defined (as the number of neurons), but in other legends not. This should be clarified.

Referee #2:

The present manuscript is in principal a follow-up to a study previously published in EMBO Reports, wherein the authors for the first time made a link between activity-dependent microRNA (miR-376) regulation and long-term-potentiation at inhibitory synapses (iLTP). In the new ms, the authors extend their studies to the canonical inhibitory synaptic scaffold gephyrin and identify miR-153 as a microRNA targeting the gephyrin 3UTR. They go on to show that miR153 is downregulated by their in vitro iLTP protocol and that ectopic expression of miR153 disrupts clustering of gephyrin and GABA-A-R at inhibitory synapses, inhibitory miniature postsynaptic currents (mIPSC) and iLTP (based on time-lapse imaging of endogenous gephyrin synaptic clusters). Moreover, they provide experimental evidence that miR153 and miR376 downregulation during iLTP is due to transcriptional repression controlled by a common calcineurin-NFAT signaling pathway. The topic of microRNA regulation of inhibitory synapses is still relatively unexplored, and the present study therefore provides a significant advance for our understanding of the mechanisms governing plasticity of the inhibitory system. Overall, I found that the presented data was mostly convincing and largely supported the claims made by the authors. However, before publication, some issues need to be addressed, including the presentation and statistical analysis of some of the data.

Major concerns:

1. The data presentation is not always state-of-the-art. Whenever possible, all individual data points should be shown (e.g.,

missing in Figs 1, 4, 6), together with a clear description of N (e.g., independent experiments, technical replicates, etc.) and the variance (preferentially not S.E.M.).

2. The statistical assessment of the data is not always appropriate. For example, for morphological (e.g., Fig. 2, 5, 6) and ephys data (Fig. 3), it appears that data points (cells) originating from different batches of neuron preparations were simply lumped together. It should be clearly stated from how many different preparations the cells originate. Most importantly, linear/mixed effects models should be used for the statistical assessment of such datasets (cf. Yu et al., *Neuron* 2022; <https://doi.org/10.1016/j.neuron.2021.10.030>). Simple T-tests/Mann-Whitney tests are not sufficient here. It should also be stated how normality of the data was assessed, which determined the use of the different tests.

3. At the moment, the functional analysis of miR-153 in iLTP completely rests on the use of miR-153 overexpression, which can be problematic due to the unphysiologically high levels of miR-153 which might be reached with the AAV constructs. Therefore, it would be important if the authors also assessed the role of inhibition of endogenous miR-153 (e.g., using sponges or antisense oligonucleotides), which should mimic iLTP in the absence of stimulation if the model presented by the authors is correct. Readouts could involve synaptic gephyrin/GABA-R clustering and/or mIPSC recordings.

4. If the authors want to make a strong point about a local preference for the miR-153/Gephyrin interaction in dendrites (which is brought up in the discussion), then they would have to use some sort of local protein synthesis assay (e.g., puro-PLA) to assess the effect of miR-153 manipulation on nascent gephyrin protein synthesis.

More minor concerns:

1. Figure 3: why is amplitude not affected, given the highly significant effect of miR153 OE on cluster area (Fig. 2)?

2. Live imaging experiments in Fig. 5 elegantly demonstrate a role for miR-153 in preventing gephyrin/GABA-R cluster growth during iLTP. This experiment should ideally be complemented with an ephys experiment to show that miR153 OE similarly prevents inhibitory iLTP.

3. The rapid reduction in pri-miR-153 expression upon iLTP is consistent with a role for transcriptional repression, but it unlikely fully explains the rapid and pronounced downregulation of mature miR-153 expression which is needed for the upregulation of gephyrin synthesis. Therefore, the authors should discuss an involvement of alternative mechanisms, e.g. activity-dependent degradation of mature microRNAs which has been previously reported (e.g., Krol et al., *Cell*; doi: 10.1016/j.cell.2010.03.039).

Referee #3:

In this study, the authors identified miR153 as a GPHN mRNA regulator. They found that during iLTP, miR153 transcripts were reduced, relieving the suppression of GPHN mRNA translation and increasing GABAergic synaptic clustering and transmission. In addition, they found the involvement of calcineurin, NFATc3, and HDACs in regulating miR153 expression using pharmacological approaches. Although this study provides some insights of the activity-dependent gene regulation contributing to GABAergic synaptic function, evidence for miR153 roles of in vivo context have not been provided. Furthermore, I suppose this study is conceptually incremental given the previous work from the same group (*Cell Rep* 2020) showing miRNA-mediated GABAR translation during iLTP. Apart from the conceptual issue, there are several concerns (see below):

1. The authors provided the evidence showing miR153 is well-expressed in cultured neurons. One way to validate the expression of miR153 is to perform quantitative immunoblots using antibodies against the previously known targets, such as VAMP2 and SNAP25.

2. The authors claimed that the number and size of inhibitory synapses were affected by miR153 by confocal imaging and live imaging. However, the analysis of colocalized puncta between presynaptic and postsynaptic markers should be presented to claim this notion. This is also true for the analysis of somatic inhibitory synapses.

3. For ex vivo electrophysiology, the authors injected AAVs into CA1 and recorded GFP-expressing pyramidal neurons. AAVs supposedly express miR153 both in pyramidal neurons and GABAergic interneurons in the CA1 area. As miR153 has been also known to affect presynaptic functions, the results obtained from Figures 3E-H cannot be solely attributed by postsynaptic GPHN regulation. I also noted that throughout the paper, most data were from cultured neurons except Figure 3E-H. Thus, I strongly suggest the authors to design experiments adequately to demonstrate the major findings in vivo context (using cell type-specific expression system (e.g. FLEX-AAVs, in conjunction with CaMKIIalpha-Cre driver lines, or sparse viral infections). It would be better to show GPHN expression by immunoblotting and inhibitory synaptic puncta by confocal microscope under the same genetic manipulations.

4. In Figure 1F, the blots for GPHN binding proteins (neuroligin-2, collybistin, IQSEC3..) should be provided.

5. For images containing immunoblot data, size markers should be accompanied.

Cross-comments by referee 1:

I have read the reviews from the other reviewers, and I agree that reviewer 3 is asking for quite a lot of additional experimental work. Particularly point 3 is proposing extensive in vivo experiments. The performed ex vivo electrophysiological recordings in CA1 neurons in the intact hippocampal circuit (presented in Figure 3) are very strong evidence supporting the main conclusions of this manuscript. The reviewer is right that potentially these effects can in part be explained by presynaptic effects of miRNA153 overexpression in CA1 interneurons. I could actually not find any information in the manuscript about whether a cell-

specific promoter was used to target expression specifically in excitatory neurons or not. Nonetheless, I would argue that clarifying and discussing this point would resolve this. This study primarily focuses on the molecular mechanistic underpinnings of iLTP and I think that additional in vivo experiments would not add significant value to the main claims of this study and would be beyond the scope of this manuscript.

Cross-comments by referee 2:

I think point 3 (in vivo physiology) is a bit beyond the scope of the study. There is still very little known about mechanisms of inhibitory plasticity, and this study provides a nice example with rigorous in vitro data. The remaining points of this reviewer however could be addressed.

Response to reviewer comments.

We thank all reviewers for their constructive comments and suggestions.

Referee #1:

In this manuscript by Welle et al., a novel miRNA-regulated pathway is described that mediates long-term potentiation (iLTP) of inhibitory synapses. In previous work (Rajgor et al., Cell Reports 2020), this group showed that iLTP is dependent on translation of synaptic proteins and showed that miRNA376c controls translation of specific GABA receptor subunits. Here, using diverse novel molecular tools, biochemistry and microscopy approaches, Welle et al., identify miR153 as a regulator of the translation of Gephyrin, the scaffold protein at inhibitory synapses that is critical for GABA receptor clustering. Specifically, they demonstrate that iLTP induction leads to the gradual downregulation of miR153 expression. Using luciferase reporter assays in several cellular systems, it is then shown that miR153 is able to downregulate Gephyrin mRNA translation by binding to the 3'UTR of Gephyrin mRNA. Consistently, overexpression of miR153 expression reduced protein levels of Gephyrin, but not of other synaptic proteins, and decreased synaptic clustering of Gephyrin and inhibitory synaptic transmission. Furthermore, it is shown that iLTP induction leads to the rapid transcriptional repression of miR153 expression that is controlled by calcineurin, NFATc3 and HDACs. Live-imaging experiments show that the gradual increase in Gephyrin and VGAT synaptic clusters after iLTP induction is abrogated in miR153-overexpression neurons, as well as in neurons treated with CaN blockers. These experiments thus indicate that iLTP leads to the reduced expression of miR153 through CaN/NFATc3/HDAC signaling, effectively relieving the translational repression of Gephyrin mRNA and promoting the translation of Gephyrin and clustering at synaptic sites - ultimately to potentiate inhibitory synaptic transmission.

Altogether, this is a very interesting, complete and rigorous study demonstrating a novel regulatory pathway underlying the regulation of iLTP. The experiments are well-controlled, properly analyzed and clearly presented. The manuscript is clearly written and has the appropriate length and format. This work is likely of broad interest for biologists, bringing important new mechanistic insight in a process that is relevant for brain functions such as learning and memory.

Comments:

- 1.1) All the experiments are consistent with the model that miR153 downregulation is required for iLTP. This is mostly based on confocal microscopy data that indeed miR153 overexpression prevents the iLTP-induced increase in the size and number of inhibitory synapses. Physiological evidence that preventing miR153 downregulation prevents potentiation of inhibitory synapses would strengthen the manuscript. For instance, a simple prediction would be that overexpression of miR153 prevents the potentiation of mIPSC frequency.
- 1.1) As suggested, we have performed electrophysiological recordings to measure mIPSCs during iLTP in cultured hippocampal neurons expressing miR153 overexpression (OE) or miRCon. These new data show that potentiation of mIPSC frequency is prevented in miR153 OE neurons. These findings concur with our imaging data which show that miR153 OE neurons do not exhibit increased GPHN and GABA_AR synaptic clustering following iLTP stimulation, and bolster the hypothesis that preventing miR153 downregulation is sufficient to disrupt GABAergic synaptic plasticity. These data are presented in the new Figure 6.
- 1.2) There is a striking difference in mIPSC frequency between the dissociated cultures and ex vivo slices. This could be very well explained by a difference in the number of inhibitory

neurons in the different preparations. Nevertheless, the kinetics of mIPSCs quantified in Sup Figure 3 are also very different. How do the authors explain this?

1.2) We anticipate that mIPSC kinetics recorded in *ex vivo* hippocampal slices are different to those in hippocampal cultures due to several factors. The mix of different neuron types found in hippocampal culture will likely provide variation in mIPSC kinetics due to the different types of GABA_ARs expressed in each cell type, which exhibit significantly different kinetic properties.^{1,2} Although we only recorded from cells with pyramidal morphologies, dissociated hippocampal cultures contain principal cells from all sub-regions of hippocampus, and therefore will express different GABA_AR compositions and exhibit a wide variety of mIPSC profiles. This is in comparison to recordings in hippocampal slices, which were restricted to pyramidal cells in the *stratum pyramidale*.

In addition to diversity of GABA_AR sub-types, mIPSC time-course is strongly determined by the concentration of GABA in the synaptic cleft, which can be impacted by multiple factors, including synapse morphology, rates of GABA clearance and myriad presynaptic factors.^{1,3} Therefore, differences in mIPSC kinetics will also be due to the differences in the local synaptic environment in intact slices compared with the dissociated culture system. These differences include the compact nature of brain slices (which would alter synapse geometry) and the substantial presence of glia (key regulators of GABA clearance).

1.3) It would be helpful if the description in the methods section for the luciferase and ChIP procedures were a bit more elaborate. How many cells are used, buffers, instruments etc?

1.3) We have now elaborated on these experimental protocols with greater detail in the methods section.

1.4) In the methods section: 'Acetyl-histone H3 chromatin immunoprecipitation' should be 'chromatin immunoprecipitation'.

1.4) We have updated the methods section with this more accurate subsection heading.

1.5) In some figure legends 'n' is defined (as the number of neurons), but in other legends not. This should be clarified.

1.5) All figure legends now include "N" in reference to the number of neuronal preparations analyzed and "n" for the number of neurons included in the analysis (where appropriate).

Referee #2:

The present manuscript is in principal a follow-up to a study previously published in EMBO Reports, wherein the authors for the first time made a link between activity-dependent microRNA (miR-376) regulation and long-term-potential at inhibitory synapses (iLTP). In the new ms, the authors extend their studies to the canonical inhibitory synaptic scaffold gephyrin and identify miR-153 as a microRNA targeting the gephyrin 3'UTR. They go on to show that miR153 is downregulated by their *in vitro* iLTP protocol and that ectopic expression of miR153 disrupts clustering of gephyrin and GABA-A-R at inhibitory synapses, inhibitory miniature postsynaptic currents (mIPSC) and iLTP (based on time-lapse imaging of endogenous gephyrin synaptic clusters). Moreover, they provide experimental evidence that miR153 and miR376 downregulation during iLTP is due to transcriptional repression controlled by a common calcineurin-NFAT signaling pathway. The topic of microRNA regulation of inhibitory synapses is still relatively unexplored, and the present study therefore provides a significant advance for our understanding of the mechanisms governing plasticity of the inhibitory system. Overall, I found

that the presented data was mostly convincing and largely supported the claims made by the authors. However, before publication, some issues need to be addressed, including the presentation and statistical analysis of some of the data.

Major concerns:

2.1) The data presentation is not always state-of-the-art. Whenever possible, all individual data points should be shown (e.g., missing in Figs 1, 4, 6), together with a clear description of N (e.g., independent experiments, technical replicates, etc.) and the variance (preferentially not S.E.M.).

2.1) In addition to updating all figure legends to clarify “N” neuronal preparations and “n” neurons (see response to point 1.5), we have also included individual data points in all figures and added more precise details to the figure legends and methods section.

2.2) The statistical assessment of the data is not always appropriate. For example, for morphological (e.g., Fig. 2, 5, 6) and ephys data (Fig. 3), it appears that data points (cells) originating from different batches of neuron preparations were simply lumped together. It should be clearly stated from how many different preparations the cells originate. Most importantly, linear/mixed effects models should be used for the statistical assessment of such datasets (cf. Yu et al., Neuron 2022; <https://doi.org/10.1016/j.neuron.2021.10.030>). Simple T-tests/Mann-Whitney tests are not sufficient here. It should also be stated how normality of the data was assessed, which determined the use of the different tests.

2.2) We have clarified the number of neuronal preparations and the number of neurons included in each analysis (see above). We have updated the methods section with extensive details regarding how normality was assessed and analysis of clustered data. Briefly, raw value data (Fig. 2, 3, 5, EV2, EV3) were assessed for normality using the D’Agostino-Pearson test or the Shapiro-Wilk test for small sample sizes ($n < 7$). For experiments in which multiple neurons (“n”) from a single neuronal culture (“N”) were analyzed (Fig. 2, 3, 5, EV2, EV3, EV5), data were organized into clusters based on their N. In such experiments comparing two groups (Fig. 2, 3, EV2, EV3), normal clustered data were analyzed with the nested t-test. Quantification graphs which include unpaired clustered data include individual data points (neurons) represented as different symbols based on neuronal preparation. Live imaging experiments entailed repeated measures of the same neurons over time (Fig. 5, 7, EV5). Statistical significance of these paired data was determined with mixed effects analysis and post-hoc multiple comparison tests.

2.3) At the moment, the functional analysis of miR-153 in iLTP completely rests on the use of miR-153 overexpression, which can be problematic due to the unphysiologically high levels of miR-153 which might be reached with the AAV constructs. Therefore, it would be important if the authors also assessed the role of inhibition of endogenous miR-153 (e.g., using sponges or antisense oligonucleotides), which should mimic iLTP in the absence of stimulation if the model presented by the authors is correct. Readouts could involve synaptic gephyrin/GABA-R clustering and/or mIPSC recordings.

2.3) We have now performed the suggested experiment, using ICC to label GPHN and VGAT in cultures expressing a miR153 inhibitor or control. Intriguingly, these experiments revealed no increase in GPHN synaptic clustering upon miR153 inhibition (Fig. EV2), which is at odds with the increased total gephyrin levels we observe upon miR153 inhibition (Fig. 1). We interpret these results in the following way: miR153 inhibition is sufficient to increase *total* protein levels of gephyrin; however, this increase is not likely not sufficient to induce the clustering of the excess gephyrin produced, without the upregulation of key proteins that

are essential for gephyrin clustering. These proteins include (but are not limited to) the GABA_ARs themselves,^{4,6} the adhesion molecule neuroligin-2,⁷ and the signaling protein, collybistin,⁷ which are not altered upon miR153 inhibition (Fig. EV1). We have also observed a similar result for the GABA_AR- α 1 and γ 2 subunit-targeting miRNA, miR376c. miR376c inhibition is not sufficient for GABA_AR clustering at synapses, as this process also requires the concurrent upregulation of gephyrin and other crucial GABA_AR subunits.^{8,9}

2.4) If the authors want to make a strong point about a local preference for the miR-153/Gephyrin interaction in dendrites (which is brought up in the discussion), then they would have to use some sort of local protein synthesis assay (e.g., puro-PLA) to assess the effect of miR-153 manipulation on nascent gephyrin protein synthesis.

2.4) While we speculate that local *Gphn* translation may explain the compartment-specific effects of miR153, it is only one possibility of many potential explanations. Further, while we have begun testing this hypothesis with puro-PLA, some studies have suggested limitations of this technique for precise localization of active translation *in situ*.^{10,11} Thus, we are currently exploring additional/alternative methods by which we could address this question. We have updated the discussion section to include these details.

More minor concerns:

2.5) Figure 3: why is amplitude not affected, given the highly significant effect of miR153 OE on cluster area (Fig. 2)?

2.5) The results of our imaging experiments show that miR153 OE disrupts inhibitory synaptic clustering in dendrites while somatic inhibitory synapses are unaffected (Fig. 2, EV2). The decreased mIPSC frequency we observe likely reflects this loss of functional dendritic synapses. However, given that somatic synapses produce higher amplitude mIPSCs than their dendritic counterparts, they have an outsized contribution to whole-cell current measurements recorded at the soma. We think that any decreased mIPSC amplitudes of dendritic synapses may be obscured by dendritic filtering and fall below our threshold for detection, and any statistically significant changes in mIPSC amplitude are being masked by unaltered, and much larger somatic mIPSCs. We observed a similar result in our previous electrophysiology studies of miRNA-mediated effects on inhibitory synapses,¹² and have now elaborated on this explanation in the manuscript.

2.6) Live imaging experiments in Fig. 5 elegantly demonstrate a role for miR-153 in preventing gephyrin/GABA-R cluster growth during iLTP. This experiment should ideally be complemented with an ephys experiment to show that miR153 OE similarly prevents inhibitory iLTP.

2.6) We have updated the manuscript to include electrophysiology experiments measuring mIPSCs in neurons overexpressing miRCon/miR153 constructs in sham or iLTP conditions. These experiments have been added to new Figure 6. Please refer to response 1.1 for more details regarding this revision.

2.7) The rapid reduction in pri-miR-153 expression upon iLTP is consistent with a role for transcriptional repression, but it unlikely fully explains the rapid and pronounced downregulation of mature miR-153 expression which is needed for the upregulation of gephyrin synthesis. Therefore, the authors should discuss an involvement of alternative mechanisms, e.g, activity-dependent degradation of mature microRNAs which has been previously reported (e.g., Krol et al., *Cell*; doi: 10.1016/j.cell.2010.03.039).

2.7) Our data reveal a rapid decrease in the primary miR153 transcript following iLTP stimulation (Fig. 4A), as well as a relatively short half-life for miR153 under basal conditions (Fig. 4B), suggesting that reduced transcription is likely a primary factor that contributes to the down regulation of mature miR153. However, as the reviewer has correctly observed, given the time-frame of such a pronounced reduction in total miR153 levels, we cannot rule out the possibility of additional mechanisms driving decreased miR153 expression, such as activity-dependent miRNA degradation previously observed in neural tissue.¹³ Error! Bookmark not defined. We have now updated the results/discussion section to address this point, including the fact that transcriptional repression may not fully account for the rapid decrease in miR153 expression observed following iLTP stimulation, and elaborating on the possibility of additional mechanisms, such as activity-dependent miRNA degradation (see page 10).

Referee #3:

In this study, the authors identified miR153 as a GPHN mRNA regulator. They found that during iLTP, miR153 transcripts were reduced, relieving the suppression of GPHN mRNA translation and increasing GABAergic synaptic clustering and transmission. In addition, they found the involvement of calcineurin, NFATc3, and HDACs in regulating miR153 expression using pharmacological approaches. Although this study provides some insights of the activity-dependent gene regulation contributing to GABAergic synaptic function, evidence for miR153 roles of in vivo context have not been provided. Furthermore, I suppose this study is conceptually incremental given the previous work from the same group (Cell Rep 2020) showing miRNA-mediated GABAR translation during iLTP. Apart from the conceptual issue, there are several concerns (see below).

We appreciate the insights and constructive comments of the reviewer. However, we respectfully disagree with the evaluation of this work as incremental in conceptual significance and our reasoning is outlined below:

- i) In this manuscript, we identify miR153 as a novel regulator of *Gphn* translation and show that its expression can directly and specifically control total GPHN protein levels and its synaptic clustering. Although *Gphn* interacts with certain RNA-binding proteins, only one of these has actually been determined as a regulator of post-transcriptional *Gphn* expression,¹⁴ and even less is known about the role of microRNAs in controlling gephyrin translation in the context of inhibitory synaptic transmission. Thus, this work addresses a gap in our knowledge of various mechanisms which control gephyrin expression in neurons.
- ii) We demonstrate that the miR153-*Gphn* interaction impacts GABA_AR synaptic clustering, GABAergic synaptic function, and plasticity. While miR153 has been shown to modulate expression of presynaptic genes,¹⁵ it is understudied in the context of the postsynaptic compartment. Further, miR153 dysregulation is implicated in Alzheimer's disease and autism spectrum disorders, which are frequently characterized by altered synaptic inhibition and excitability.¹⁶⁻²⁵ Identifying miR153 as a crucial regulator of inhibitory synaptic structure, function, and plasticity represents a key finding for understanding how miRNA systems can contribute to these pathologies.
- iii) As the reviewer notes, we have previously observed a similar mechanism during iLTP, in which a different miRNA (miR376c) mediates translation of synaptic GABA_AR subunits (α 1 and γ 2).¹² Here we discover that, like miR376c, miR153 transcription is rapidly downregulated following iLTP stimulation to allow increased translation of its target, and this transcriptional repression requires Ca²⁺/CaN signaling. While our findings here parallel what we have observed previously with GABA_ARs and miR376c, we do not think this diminishes impact, but enhances it. We have uncovered a common E-T coupling pathway

which leverages the expression of a network of miRNAs to alter translation of multiple target genes in response to activity. Thus, our work contributes to our understanding of inhibitory synaptic transmission and lays the groundwork for future research into activity-dependent gene regulation and the scope of its role in inhibitory synaptic plasticity.

- 3.1) The authors provided the evidence showing miR153 is well-expressed in cultured neurons. One way to validate the expression of miR153 is to perform quantitative immunoblots using antibodies against the previously known targets, such as VAMP2 and SNAP25.
- 3.1) We have now performed immunoblotting experiments probing for expression of VAMP2 in cultures infected with miR153 OE construct, to validate its expression. We find that VAMP2 expression is decreased in miR153 OE neurons, and these results have been added to Figure EV1.
- 3.2) The authors claimed that the number and size of inhibitory synapses were affected by miR153 by confocal imaging and live imaging. However, the analysis of colocalized puncta between presynaptic and postsynaptic markers should be presented to claim this notion. This is also true for the analysis of somatic inhibitory synapses.
- 3.2) We have performed new analyses of the immunocytochemical data to assess the difference in cluster area and density for GPHN or GABA_AR puncta which are co-localized with presynaptic VGAT puncta. These results are added to Figures 2 and EV2.
- 3.3) For ex vivo electrophysiology, the authors injected AAVs into CA1 and recorded GFP-expressing pyramidal neurons. AAVs supposedly express miR153 both in pyramidal neurons and GABAergic interneurons in the CA1 area. As miR153 has been also known to affect presynaptic functions, the results obtained from Figures 3E-H cannot be solely attributed by postsynaptic GPHN regulation. I also noted that throughout the paper, most data were from cultured neurons except Figure 3E-H. Thus, I strongly suggest the authors to design experiments adequately to demonstrate the major findings in vivo context (using cell type-specific expression system (e.g. FLEX-AAVs, in conjunction with CaMKIIalpha-Cre driver lines, or sparse viral infections). It would be better to show GPHN expression by immunoblotting and inhibitory synaptic puncta by confocal microscope under the same genetic manipulations.
- 3.3) We understand these questions and concerns. Indeed, we cannot eliminate the possibility of presynaptic effects in our *ex vivo* recording experiments, given the nature of the viral infections and the fact that miR153 also targets genes involved in presynaptic functions. However, our *in vitro* experiments utilize sparse transfections of the miR153 OE construct, isolating the impacts of miR153 overexpression in the postsynaptic neuron. Both our *in vitro* and *ex vivo* experiments demonstrate a reduction in mIPSC frequency in miR153 OE neurons (Fig. 3). This effect, in combination with the imaging data which show disruption of postsynaptic GPHN and GABA_AR clusters in sparsely transfected miR153 OE neurons (Fig. 2), suggests that the decreased mIPSC frequency we observe *ex vivo* is not likely due solely to presynaptic changes. We have now discussed this caveat in more detail in the results and discussion sections of the manuscript. Given the amount of work it would take for our small lab to repeat these experiments *in vivo*, in a cell-specific expression system, we agree with the other reviewers' assessment that this represents a future direction, which falls beyond the scope of our current study.
- 3.4) In Figure 1F, the blots for GPHN binding proteins (neuroligin-2, collybistin, IQSEC3..) should be provided.

3.4) We have now included immunoblots probing for the GPHN-binding proteins neuroligin-2 (NL2) and collybistin (CB) in neurons overexpressing miR153. The results of these experiments, included in Figure EV1, show there is no significant difference in NL2 or CB expression when miR153 is overexpressed. This indicates that modulating miR153 expression specifically impacts GPHN levels without affecting the expression of its interacting proteins.

3.5) For images containing immunoblot data, size markers should be accompanied.

3.5) Immunoblot images in Figures 1, 7 and Figure EV1 now include size references.

Cross comments

Referee #1:

Cross-comments: I have read the reviews from the other reviewers, and I agree that reviewer 3 is asking for quite a lot of additional experimental work. Particularly point 3 is proposing extensive *in vivo* experiments. The performed *ex vivo* electrophysiological recordings in CA1 neurons in the intact hippocampal circuit (presented in Figure 3) are very strong evidence supporting the main conclusions of this manuscript. The reviewer is right that potentially these effects can in part be explained by presynaptic effects of miRNA153 overexpression in CA1 interneurons. I could actually not find any information in the manuscript about whether a cell-specific promoter was used to target expression specifically in excitatory neurons or not. Nonetheless, I would argue that clarifying and discussing this point would resolve this. This study primarily focuses on the molecular mechanistic underpinnings of iLTP and I think that additional *in vivo* experiments would not add significant value to the main claims of this study and would be beyond the scope of this manuscript.

We have now clarified in the results/methods sections that we did not utilize cell-specific promoters and elaborated on the potential role of presynaptic effects in the results/discussion section.

Referee #2:

Cross-comments: I think point 3 (*in vivo* physiology) is a bit beyond the scope of the study. There is still very little known about mechanisms of inhibitory plasticity, and this study provides a nice example with rigorous *in vitro* data. The remaining points of this reviewer however could be addressed.

Thank you for your cross-commentary. We have, as suggested, addressed the concerns of Referee #3 (with the exception of *in vivo* experiments) throughout the manuscript as detailed in responses 3.1-3.5.

References

-
- 1) Barberis A, Petrini EM, Mozrzymas JW. Impact of synaptic neurotransmitter concentration time course on the kinetics and pharmacological modulation of inhibitory synaptic currents. *Front Cell Neurosci*. 2011 Jun 22;5:6. doi: 10.3389/fncel.2011.00006. PMID: 21734864; PMCID: PMC3123770.
 - 2) Eyre MD, Renzi M, Farrant M, Nusser Z. Setting the time course of inhibitory synaptic currents by mixing multiple GABA(A) receptor α subunit isoforms. *J Neurosci*. 2012 Apr 25;32(17):5853-67. doi: 10.1523/JNEUROSCI.6495-11.2012. PMID: 22539847; PMCID: PMC3348502.
 - 3) Cherubini E, Conti F. Generating diversity at GABAergic synapses. *Trends Neurosci*. 2001 Mar;24(3):155-62. doi: 10.1016/s0166-2236(00)01724-0. PMID: 11182455.

-
- 4) Essrich, C., Lorez, M., Benson, J.A., Fritschy, J.-M., and Lüscher, B. (1998). Postsynaptic clustering of major GABA_A receptor subtypes requires the $\alpha 2$ subunit and gephyrin. *Nat. Neurosci.* 1, 563–571.
 - 5) Panzanelli, P., Gunn, B.G., Schlatter, M.C., Benke, D., Tyagarajan, S.K., Scheiffele, P., Belelli, D., Lambert, J.J., Rudolph, U., and Fritschy, J.M. (2011). Distinct mechanisms regulate GABA_A receptor and gephyrin clustering at perisomatic and axo-axonic synapses on CA1 pyramidal cells. *J. Physiol.* 589, 4959–4980
 - 6) Schweizer, C., Balsiger, S., Bluethmann, H., Mansuy, I.M., Fritschy, J.-M., Mohler, H., and Lüscher, B. (2003). The $\alpha 2$ subunit of GABA(A) receptors is required for maintenance of receptors at mature synapses. *Mol. Cell. Neurosci.* 24, 442–450.
 - 7) Papadopoulos T, Eulenburg V, Reddy-Alla S, Mansuy IM, Li Y, Betz H. Collybistin is required for both the formation and maintenance of GABAergic postsynapses in the hippocampus. *Mol Cell Neurosci.* 2008 Oct;39(2):161-9. doi: 10.1016/j.mcn.2008.06.006. Epub 2008 Jun 20. PMID: 18625319.
 - 8) Connolly, C.N., Krishek, B.J., McDonald, B.J., Smart, T.G., and Moss, S.J. (1996). Assembly and cell surface expression of heteromeric and homomeric gamma-aminobutyric acid type A receptors. *J Biol Chem* 271, 89-96.
 - 9) Hannan S, Smart TG. Cell surface expression of homomeric GABA_A receptors depends on single residues in subunit transmembrane domains. *J Biol Chem.* 2018 Aug 31;293(35):13427-13439. doi: 10.1074/jbc.RA118.002792. Epub 2018 Jul 9. PMID: 29986886; PMCID: PMC6120189.
 - 10) Hobson BD, Kong L, Hartwick EW, Gonzalez RL, Sims PA. Elongation inhibitors do not prevent the release of puromycylated nascent polypeptide chains from ribosomes. *Elife.* 2020 Aug 26;9:e60048. doi: 10.7554/eLife.60048. PMID: 32844746; PMCID: PMC7490010.
 - 11) Enam SU, Zinshteyn B, Goldman DH, Cassani M, Livingston NM, Seydoux G, Green R. Puromycin reactivity does not accurately localize translation at the subcellular level. *Elife.* 2020 Aug 26;9:e60303. doi: 10.7554/eLife.60303. PMID: 32844748; PMCID: PMC7490009.
 - 12) Rajgor D, Purkey AM, Sanderson JL, Welle TM, Garcia JD, Dell'Acqua ML, Smith KR. Local miRNA-Dependent Translational Control of GABA_A Receptor Synthesis during Inhibitory Long-Term Potentiation. *Cell Rep.* 2020 Jun 23;31(12):107785. doi: 10.1016/j.celrep.2020.107785. PMID: 32579917; PMCID: PMC7486624.
 - 13) Krol J, Buskamp V, Markiewicz I, Stadler MB, Ribí S, Richter J, Duebel J, Bicker S, Fehling HJ, Schübeler D, Oertner TG, Schratt G, Bibel M, Roska B, Filipowicz W. Characterizing light-regulated retinal microRNAs reveals rapid turnover as a common property of neuronal microRNAs. *Cell.* 2010 May 14;141(4):618-31. doi: 10.1016/j.cell.2010.03.039. PMID: 20478254.
 - 14) Zahr SK, Yang G, Kazan H, Borrett MJ, Yuzwa SA, Voronova A, Kaplan DR, Miller FD. A Translational Repression Complex in Developing Mammalian Neural Stem Cells that Regulates Neuronal Specification. *Neuron.* 2018 Feb 7;97(3):520-537.e6. doi: 10.1016/j.neuron.2017.12.045. Epub 2018 Jan 27. PMID: 29395907.
 - 15) Mathew RS, Tatarakis A, Rudenko A, Johnson-Venkatesh EM, Yang YJ, Murphy EA, Todd TP, Schepers ST, Siuti N, Martorell AJ, Falls WA, Hammack SE, Walsh CA, Tsai LH, Umehara H, Bouton ME, Moazed D. A microRNA negative feedback loop downregulates vesicle transport and inhibits fear memory. *Elife.* 2016 Dec 21;5:e22467. doi: 10.7554/eLife.22467. PMID: 28001126; PMCID: PMC5293492.
 - 16) Yan ML, Zhang S, Zhao HM, Xia SN, Jin Z, Xu Y, Yang L, Qu Y, Huang SY, Duan MJ, Mao M, An XB, Mishra C, Zhang XY, Sun LH, Ai J. MicroRNA-153 impairs presynaptic plasticity by blocking vesicle release following chronic brain hypoperfusion. *Cell Commun Signal.* 2020 Apr 6;18(1):57. doi: 10.1186/s12964-020-00551-8. PMID: 32252776; PMCID: PMC7137307.
 - 17) You YH, Qin ZQ, Zhang HL, Yuan ZH, Yu X. MicroRNA-153 promotes brain-derived neurotrophic factor and hippocampal neuron proliferation to alleviate autism symptoms through inhibition of JAK-STAT pathway by LEPR. *Biosci Rep.* 2019 Jun 25;39(6):BSR20181904. doi: 10.1042/BSR20181904. PMID: 30975733; PMCID: PMC6591574.

-
- 18) Qiao J, Zhao J, Chang S, Sun Q, Liu N, Dong J, Chen Y, Yang D, Ye D, Liu X, Yu Y, Chen W, Zhu S, Wang G, Jia W, Xi J, Kang J. MicroRNA-153 improves the neurogenesis of neural stem cells and enhances the cognitive ability of aged mice through the notch signaling pathway. *Cell Death Differ.* 2020 Feb;27(2):808-825. doi: 10.1038/s41418-019-0388-4. Epub 2019 Jul 11. PMID: 31296962; PMCID: PMC7206122.
- 19) Gupta P, Bhattacharjee S, Sharma AR, Sharma G, Lee SS, Chakraborty C. miRNAs in Alzheimer Disease - A Therapeutic Perspective. *Curr Alzheimer Res.* 2017;14(11):1198-1206. doi: 10.2174/1567205014666170829101016. PMID: 28847283.
- 20) Bi D, Wen L, Wu Z, Shen Y. GABAergic dysfunction in excitatory and inhibitory (E/I) imbalance drives the pathogenesis of Alzheimer's disease. *Alzheimers Dement.* 2020 Sep;16(9):1312-1329. doi: 10.1002/alz.12088. Epub 2020 Jun 16. PMID: 32543726.
- 21) Busche MA, Konnerth A. Impairments of neural circuit function in Alzheimer's disease. *Philos Trans R Soc Lond B Biol Sci.* 2016 Aug 5;371(1700):20150429. doi: 10.1098/rstb.2015.0429. PMID: 27377723; PMCID: PMC4938029.
- 22) Yizhar O, Fenno LE, Prigge M, Schneider F, Davidson, TJ, O'Shea, DJ, Sohal VS, Goshen I, Finkelstein J, Paz JT, Stehfest K, Fudim R, Ramakrishnan C, Huguenard JR, Hegemann P, Deisseroth K. Neocortical excitation/inhibition balance in information processing and social dysfunction. *Nature* **477**, 171–178 (2011). <https://doi.org/10.1038/nature10360>
- 23) Ferguson BR, Gao WJ. PV Interneurons: Critical Regulators of E/I Balance for Prefrontal Cortex-Dependent Behavior and Psychiatric Disorders. *Front Neural Circuits.* 2018 May 16;12:37. doi: 10.3389/fncir.2018.00037. PMID: 29867371; PMCID: PMC5964203.
- 24) Selten M, van Bokhoven H, Nadif Kasri N. Inhibitory control of the excitatory/inhibitory balance in psychiatric disorders. *F1000Res.* 2018 Jan 8;7:23. doi: 10.12688/f1000research.12155.1. PMID: 29375819; PMCID: PMC5760969.
- 25) Gao R, Penzes P. Common mechanisms of excitatory and inhibitory imbalance in schizophrenia and autism spectrum disorders. *Curr Mol Med.* 2015;15(2):146-67. doi: 10.2174/1566524015666150303003028. PMID: 25732149; PMCID: PMC4721588.

Dear Dr. Smith

Thank you for the submission of your revised manuscript to EMBO Reports. Since my colleague Esther Schnapp is currently out of office, I have stepped in as the secondary editor for your manuscript.

We have now received the full set of referee reports that is copied below, and as you will see all three referees are very positive about the study and recommend publication.

Browsing through the manuscript myself, I noticed a few editorial things that we need before we can proceed with the official acceptance of your study.

- Please provide up to 5 keywords
- Please remove the DOIs from the reference list. These are only needed for preprints and datasets that have not been published yet.
- Funding information: the following grant is provided in the manuscript as 5T32NS099042 , while it is entered as T32NS099042 in the online manuscript tracking system (the '5' is missing). Please check which version is the correct one.
- Please provide the EV Figures as individual production quality figure files (.eps, .tif, .jpg; one file per figure).
- All figure callouts that have the prefix "S" need to be updated/corrected since your manuscript does not contain "S" figures (e.g. Fig. S2A-B,E). I assume these are referring to EV figures for which the nomenclature is Figure EV2A-B, E)?
- Since July 1st we require a Reagent and Tools table in the Methods section, listing key reagents, experimental models, software and relevant equipment and including their sources and relevant identifiers. A downloadable template (.docx) for the Reagents and Tools Table can be found in our author guidelines:
<https://www.embopress.org/page/journal/14693178/authorguide#structuredmethods>.

An example of a Method paper with Structured Methods can be found here:
<https://www.embopress.org/doi/10.15252/msb.20178071>.

- Experimental animals: please provide the reference number for approval in the methods section.
- MATERIALS & METHODS should be METHODS
- Our production/data editors have asked you to clarify several points in the figure legends (see below). Please incorporate these changes in the manuscript and return the revised file with tracked changes with your final manuscript submission.

A) Statistical test information. Only p-values that are actually shown in the figure panel(s) should (and must) be defined in the legends, all others should be removed from (or added to) the legend. Moreover, we ask for the specification of exact p-values:

- Please note that the exact p values are not provided in the legends of figures 1b, d-e, g-h; 2b-c, e-f; 3b, f; 4a-d, f-j; 5b-c, e; 6b; 7a, c, e-f; EV 1b, d; EV 4c; EV 5b-f.
- Please note that in figures 1b, d-e, g-h; 2b-c, e-f; 3b, f; 4a-d, f-j; 6b; EV 1b-e; EV 2b, d-g, i, k; EV 3a-b; EV 4c; EV 5b-f, there is a mismatch between the annotated p values in the figure legend and the annotated p values in the figure file that should be corrected.

B) Replicates and error bars:

- Although 'n' is provided, please describe the nature of entity for 'n' in the legends of figures 1b, d-e, g-h; 7a, c; EV 1b, d; EV 4c.
- Finally, EMBO Reports papers are accompanied online by a schematic summary figure that provides a sketch of the major findings (synopsis image). Please provide this figure as a separate file in PNG or JPG format at a size of 550x300-600 pixels (width x height). Please note that the size is rather small and that text needs to be readable at the final size.

With kind regards,

Martina Rembold, PhD
Senior Editor

EMBO reports

Referee #1:

The authors provide substantial additional experimental work supporting the main conclusion of the study and adequately revised the manuscript text. This is now a very complete and rigorous study, providing important new mechanistic insight in the molecular processes that underlie plasticity of inhibitory synapses. I regard this manuscript as suitable for publication in EMBO Reports.

Referee #2:

The authors have satisfactorily addressed all my previous concerns, and I can recommend publication of this paper in EMBO Reports without reservation. This is the second report from this group demonstrating an involvement of the microRNA pathway in the regulation of long-term potentiation at inhibitory synapses, a fairly understudied process. This should have important implications for our understanding of neural circuit function and memory-related processes in health and disease.

Referee #3:

The authors have effectively addressed my concerns. However, I still believe that ex-vivo recording is necessary utilizing a cell-type-specific expression system to differentiate the impact of miR153 on pre- and postsynaptic functions, given that the well-known target of miR153 is VAMP2. Nonetheless, I am content with the way this issue has been discussed in the Discussion section. I look forward to future studies clarifying this issue.

All editorial and formatting issues were resolved by the authors.

Dr. Katharine Smith
University of Colorado Denver School of Medicine
Pharmacology
Anschutz Medical Campus
12000 E. 19th Avenue
Aurora, CO 80045
United States

Dear Dr. Smith,

I am very pleased to accept your manuscript for publication in the next available issue of EMBO reports. Thank you for your contribution to our journal.
